# Simple and Effective Regularization Methods for Training on Noisily Labeled Data with Generalization Guarantee

**Wei Hu**    **Zhiyuan Li**    **Dingli Yu**
Princeton University
{huwei,zhiyuanli,dingliy}@cs.princeton.edu

## Abstract

Over-parameterized deep neural networks trained by simple first-order methods are known to be able to fit any labeling of data. Such over-fitting ability hinders generalization when mislabeled training examples are present. On the other hand, simple regularization methods like early-stopping can often achieve highly nontrivial performance on clean test data in these scenarios, a phenomenon not theoretically understood. This paper proposes and analyzes two simple and intuitive regularization methods: (i) regularization by the distance between the network parameters to initialization, and (ii) adding a trainable auxiliary variable to the network output for each training example. Theoretically, we prove that gradient descent training with either of these two methods leads to a generalization guarantee on the clean data distribution despite being trained using noisy labels. Our generalization analysis relies on the connection between wide neural network and neural tangent kernel (NTK). The generalization bound is independent of the network size, and is comparable to the bound one can get when there is no label noise. Experimental results verify the effectiveness of these methods on noisily labeled datasets.

## 1 Introduction

Modern deep neural networks are trained in a highly over-parameterized regime, with many more trainable parameters than training examples. It is well-known that these networks trained with simple first-order methods can fit any labels, even completely random ones (Zhang et al., 2017). Although training on properly labeled data usually leads to good generalization performance, the ability to over-fit the entire training dataset is undesirable for generalization when noisy labels are present. Therefore preventing over-fitting is crucial for robust performance since mislabeled data are ubiquitous in very large datasets (Krishna et al., 2016).

In order to prevent over-fitting to mislabeled data, some form of *regularization* is necessary. A simple such example is *early stopping*, which has been observed to be surprisingly effective for this purpose (Rolnick et al., 2017; Guan et al., 2018; Li et al., 2019). For instance, training ResNet-34 with early stopping can achieve $84\%$ test accuracy on CIFAR-10 even when $60\%$ of the training labels are corrupted (Table 1). This is nontrivial since the test error is much smaller than the error rate in training data. How to explain such generalization phenomenon is an intriguing theoretical question.

As a step towards a theoretical understanding of the generalization phenomenon for over-parameterized neural networks when noisy labels are present, this paper proposes and analyzes two simple regularization methods as alternatives of early stopping:

1. Regularization by *distance to initialization*. Denote by $\boldsymbol{\theta}$ the network parameters and by $\boldsymbol{\theta}(0)$ its random initialization. This method adds a regularizer $\lambda \|\boldsymbol{\theta} - \boldsymbol{\theta}(0)\|^2$ to the training objective.

2. Adding an *auxiliary variable* for each training example. Let $\boldsymbol{x}_i$ be the $i$-th training example and $\boldsymbol{f}(\boldsymbol{\theta}, \cdot)$ represent the neural net. This method adds a trainable variable $\boldsymbol{b}_i$ and tries to fit the $i$-th label using $\boldsymbol{f}(\boldsymbol{\theta}, \boldsymbol{x}_i) + \lambda \boldsymbol{b}_i$. At test time, only the neural net $\boldsymbol{f}(\boldsymbol{\theta}, \cdot)$ is used and the auxiliary variables are discarded.

These two choices of regularization are well motivated with clear intuitions. First, distance to initialization has been observed to be very related to generalization in deep learning (Neyshabur et al., 2019; Nagarajan and Kolter, 2019), so regularizing by distance to initialization can potentially help generalization. Second, the effectiveness of early stopping indicates that clean labels are somewhat easier to fit than wrong labels; therefore, adding an auxiliary variable could help "absorb" the noise in the labels, thus making the neural net itself not over-fitting.

We provide theoretical analysis of the above two regularization methods for a class of sufficiently wide neural networks by proving a generalization bound for the trained network on *clean* data distribution when the training dataset contains noisy labels. Our generalization bound depends on the (unobserved) clean labels, and is comparable to the bound one can get when there is no label noise, therefore indicating that the proposed regularization methods are robust to noisy labels.

Our theoretical analysis is based on the recently established connection between wide neural net and *neural tangent kernel* (Jacot et al., 2018; Lee et al., 2019; Arora et al., 2019a). In this line of work, parameters in a wide neural net are shown to stay close to their initialization during gradient descent training, and as a consequence, the neural net can be effectively approximated by its first-order Taylor expansion with respect to its parameters at initialization. This leads to tractable linear dynamics under $\ell_2$ loss, and the final solution can be characterized by kernel regression using a particular kernel named neural tangent kernel (NTK). In fact, we show that for wide neural nets, both of our regularization methods, when trained with gradient descent to convergence, correspond to *kernel ridge regression* using the NTK, which is often regarded as an alternative to early stopping in kernel literature. This viewpoint makes explicit the connection between our methods and early stopping.

The effectiveness of these two regularization methods is verified empirically – on MNIST and CIFAR-10, they are able to achieve highly nontrivial test accuracy, on a par with or even better than early stopping. Furthermore, with our regularization, the validation accuracy is almost monotone increasing throughout the entire training process, indicating their resistance to over-fitting.

## 2 RELATED WORK

Neural tangent kernel was first explicitly studied and named by Jacot et al. (2018), with several further refinements and extensions by Lee et al. (2019); Yang (2019); Arora et al. (2019a). Using the similar idea that weights stay close to initialization and that the neural network is approximated by a linear model, a series of theoretical papers studied the optimization and generalization issues of very wide deep neural nets trained by (stochastic) gradient descent (Du et al., 2019b; 2018b; Li and Liang, 2018; Allen-Zhu et al., 2018a;b; Zou et al., 2018; Arora et al., 2019b; Cao and Gu, 2019). Empirically, variants of NTK on convolutional neural nets and graph neural nets exhibit strong practical performance (Arora et al., 2019a; Du et al., 2019a), thus suggesting that ultra-wide (or infinitely wide) neural nets are at least not irrelevant.

Our methods are closely related to kernel ridge regression, which is one of the most common kernel methods and has been widely studied. It was shown to perform comparably to early-stopped gradient descent (Bauer et al., 2007; Gerfo et al., 2008; Raskutti et al., 2014; Wei et al., 2017). Accordingly, we indeed observe in our experiments that our regularization methods perform similarly to gradient descent with early stopping in neural net training.

In another theoretical study relevant to ours, Li et al. (2019) proved that gradient descent with early stopping is robust to label noise for an over-parameterized two-layer neural net. Under a clustering assumption on data, they showed that gradient descent fits the correct labels before starting to over-fit wrong labels. Their result is different from ours from several aspects: they only considered two-layer nets while we allow arbitrarily deep nets; they required a clustering assumption on data while our generalization bound is general and data-dependent; furthermore, they did not address the question of generalization, but only provided guarantees on the training data.

A large body of work proposed various methods for training with mislabeled examples, such as estimating noise distribution (Liu and Tao, 2015) or confusion matrix (Sukhbaatar et al., 2014), using surrogate loss functions (Ghosh et al., 2017; Zhang and Sabuncu, 2018), meta-learning (Ren et al., 2018), using a pre-trained network (Jiang et al., 2017), and training two networks simultaneously (Malach and Shalev-Shwartz, 2017; Han et al., 2018; Yu et al., 2019). While our methods are not necessarily superior to these methods in terms of performance, our methods are arguably simpler (with minimal change to normal training procedure) and come with formal generalization guarantee.

## 3 PRELIMINARIES

**Notation.** We use bold-faced letters for vectors and matrices. We use $\|\cdot\|$ to denote the Euclidean norm of a vector or the spectral norm of a matrix, and $\|\cdot\|_F$ to denote the Frobenius norm of a matrix. $\langle\cdot,\cdot\rangle$ represents the standard inner product. Let $\boldsymbol{I}$ be the identity matrix of appropriate dimension. Let $[n] = \{1, 2, \ldots, n\}$. Let $\mathbb{I}[A]$ be the indicator of event $A$.

### 3.1 SETTING: LEARNING FROM NOISILY LABELED DATA

Now we formally describe the setting considered in this paper. We first describe the binary classification setting as a warm-up, and then describe the more general setting of multi-class classification.

**Binary classification.** Suppose that there is an underlying data distribution $\mathcal{D}$ over $\mathbb{R}^d \times \{\pm 1\}$, where $1$ and $-1$ are labels corresponding to two classes. However, we only have access to samples from a noisily labeled version of $\mathcal{D}$. Formally, the data generation process is: draw $(\boldsymbol{x}, y) \sim \mathcal{D}$, and flip the sign of label $y$ with probability $p$ ($0 \leqslant p < \frac{1}{2}$); let $\tilde{y} \in \{\pm 1\}$ be the resulting noisy label.

Let $\{(\boldsymbol{x}_i, \tilde{y}_i)\}_{i=1}^n$ be i.i.d. samples generated from the above process. Although we only have access to these noisily labeled data, the goal is still to learn a function (in the form of a neural net) that can predict the true label well on the clean distribution $\mathcal{D}$. For binary classification, it suffices to learn a single-output function $f : \mathbb{R}^d \to \mathbb{R}$ whose sign is used to predict the class, and thus the classification error of $f$ on $\mathcal{D}$ is defined as $\Pr_{(\boldsymbol{x},y)\sim\mathcal{D}}[\text{sgn}(f(\boldsymbol{x})) \neq y]$.

**Multi-class classification.** When there are $K$ classes ($K > 2$), let the underlying data distribution $\mathcal{D}$ be over $\mathbb{R}^d \times [K]$. We describe the noise generation process as a matrix $\boldsymbol{P} \in \mathbb{R}^{K \times K}$, whose entry $p_{c',c}$ is the probability that the label $c$ is transformed into $c'$ ($\forall c, c' \in [K]$). Therefore the data generation process is: draw $(\boldsymbol{x}, c) \sim \mathcal{D}$, and replace the label $c$ with $\tilde{c}$ from the distribution $\Pr[\tilde{c} = c'|c] = p_{c',c}$ ($\forall c' \in [K]$).

Let $\{(\boldsymbol{x}_i, \tilde{c}_i)\}_{i=1}^n$ be i.i.d. samples from the above process. Again we would like to learn a neural net with low classification error on the clean distribution $\mathcal{D}$. For $K$-way classification, it is common to use a neural net with $K$ outputs, and the index of the maximum output is used to predict the class. Thus for $\boldsymbol{f} : \mathbb{R}^d \to \mathbb{R}^K$, its (top-1) classification error on $\mathcal{D}$ is $\Pr_{(\boldsymbol{x},c)\sim\mathcal{D}}[c \notin \arg\max_{h\in[K]} f^{(h)}(\boldsymbol{x})]$, where $f^{(h)} : \mathbb{R}^d \to \mathbb{R}$ is the function computed by the $h$-th output of $\boldsymbol{f}$.

As standard practice, a class label $c \in [K]$ is also treated as its *one-hot encoding* $\boldsymbol{e}^{(c)} = (0, 0, \cdots, 0, 1, 0, \cdots, 0) \in \mathbb{R}^K$ (the $c$-th coordinate being 1), which can be paired with the $K$ outputs of the network and fed into a loss function during training.

Note that it is necessary to assume $p_{c,c} > p_{c',c}$ for all $c \neq c'$, i.e., the probability that a class label $c$ is transformed into another particular label must be smaller than the label $c$ being correct – otherwise it is impossible to identify class $c$ correctly from noisily labeled data.

### 3.2 RECAP OF NEURAL TANGENT KERNEL

Now we briefly and informally recap the theory of neural tangent kernel (NTK) (Jacot et al., 2018; Lee et al., 2019; Arora et al., 2019a), which establishes the equivalence between training a wide neural net and a kernel method.

We first consider a neural net with a scalar output, defined as $f(\boldsymbol{\theta}, \boldsymbol{x}) \in \mathbb{R}$, where $\boldsymbol{\theta} \in \mathbb{R}^N$ is all the parameters in the net and $\boldsymbol{x} \in \mathbb{R}^d$ is the input. Suppose that the net is trained by minimizing the $\ell_2$ loss over a training dataset $\{(\boldsymbol{x}_i, y_i)\}_{i=1}^n \subset \mathbb{R}^d \times \mathbb{R}$: $L(\boldsymbol{\theta}) = \frac{1}{2} \sum_{i=1}^n (f(\boldsymbol{\theta}, \boldsymbol{x}_i) - y_i)^2$. Let the random initial parameters be $\boldsymbol{\theta}(0)$, and the parameters be updated according to gradient descent on $L(\boldsymbol{\theta})$. It is shown that if the network is sufficiently wide[1], the parameters $\boldsymbol{\theta}$ will stay close to the initialization $\boldsymbol{\theta}(0)$ during training so that the following first-order approximation is accurate:

$$f(\boldsymbol{\theta}, \boldsymbol{x}) \approx f(\boldsymbol{\theta}(0), \boldsymbol{x}) + \langle\nabla_{\boldsymbol{\theta}} f(\boldsymbol{\theta}(0), \boldsymbol{x}), \boldsymbol{\theta} - \boldsymbol{\theta}(0)\rangle. \tag{1}$$

This approximation is exact in the infinite width limit, but can also be shown when the width is finite but sufficiently large. When approximation (1) holds, we say that we are in the *NTK regime*.

Define $\phi(\boldsymbol{x}) = \nabla_{\boldsymbol{\theta}} f(\boldsymbol{\theta}(0), \boldsymbol{x})$ for any $\boldsymbol{x} \in \mathbb{R}^d$. The right hand side in (1) is linear in $\boldsymbol{\theta}$. As a consequence, training on the $\ell_2$ loss with gradient descent leads to the *kernel regression* solution

---

[1]"Width" refers to number of nodes in a fully connected layer or number of channels in a convolutional layer.

with respect to the kernel induced by (random) features $\phi(\boldsymbol{x})$, which is defined as $k(\boldsymbol{x}, \boldsymbol{x}') = \langle \phi(\boldsymbol{x}), \phi(\boldsymbol{x}') \rangle$ for $\boldsymbol{x}, \boldsymbol{x}' \in \mathbb{R}^d$. This kernel was named the neural tangent kernel (NTK) by Jacot et al. (2018). Although this kernel is random, it is shown that when the network is sufficiently wide, this random kernel converges to a deterministic limit in probability (Arora et al., 2019a). If we additionally let the neural net and its initialization be defined so that the initial output is small, i.e., $f(\boldsymbol{\theta}(0), \boldsymbol{x}) \approx 0$,[2] then the network at the end of training approximately computes the following function:

$$\boldsymbol{x} \mapsto k(\boldsymbol{x}, \boldsymbol{X})^\top \left( k(\boldsymbol{X}, \boldsymbol{X}) \right)^{-1} \boldsymbol{y}, \tag{2}$$

where $\boldsymbol{X} = (\boldsymbol{x}_1, \ldots, \boldsymbol{x}_n)$ is the training inputs, $\boldsymbol{y} = (y_1, \ldots, y_n)^\top$ is the training targets, $k(\boldsymbol{x}, \boldsymbol{X}) = (k(\boldsymbol{x}, \boldsymbol{x}_1), \ldots, k(\boldsymbol{x}, \boldsymbol{x}_n))^\top \in \mathbb{R}^n$, and $k(\boldsymbol{X}, \boldsymbol{X}) \in \mathbb{R}^{n \times n}$ with $(i, j)$-th entry being $k(\boldsymbol{x}_i, \boldsymbol{x}_j)$.

**Multiple outputs.** The NTK theory above can also be generalized straightforwardly to the case of multiple outputs (Jacot et al., 2018; Lee et al., 2019). Suppose we train a neural net with $K$ outputs, $\boldsymbol{f}(\boldsymbol{\theta}, \boldsymbol{x})$, by minimizing the $\ell_2$ loss over a training dataset $\{(\boldsymbol{x}_i, \boldsymbol{y}_i)\}_{i=1}^n \subset \mathbb{R}^d \times \mathbb{R}^K$: $L(\boldsymbol{\theta}) = \frac{1}{2} \sum_{i=1}^n \| \boldsymbol{f}(\boldsymbol{\theta}, \boldsymbol{x}_i) - \boldsymbol{y}_i \|^2$. When the hidden layers are sufficiently wide such that we are in the NTK regime, at the end of gradient descent, each output of $\boldsymbol{f}$ also attains the kernel regression solution with respect to the same NTK as before, using the corresponding dimension in the training targets $\boldsymbol{y}_i$. Namely, the $h$-th output of the network computes the function

$$f^{(h)}(\boldsymbol{x}) = k(\boldsymbol{x}, \boldsymbol{X})^\top \left( k(\boldsymbol{X}, \boldsymbol{X}) \right)^{-1} \boldsymbol{y}^{(h)},$$

where $\boldsymbol{y}^{(h)} \in \mathbb{R}^n$ whose $i$-th coordinate is the $h$-th coordinate of $\boldsymbol{y}_i$.

## 4 REGULARIZATION METHODS

In this section we describe two simple regularization methods for training with noisy labels, and show that if the network is sufficiently wide, both methods lead to kernel ridge regression using the NTK.[3]

We first consider the case of scalar target and single-output network. The generalization to multiple outputs is straightforward and is treated at the end of this section. Given a noisily labeled training dataset $\{(\boldsymbol{x}_i, \tilde{y}_i)\}_{i=1}^n \subset \mathbb{R}^d \times \mathbb{R}$, let $f(\boldsymbol{\theta}, \cdot)$ be a neural net to be trained. A direct, unregularized training method would involve minimizing an objective function like $L(\boldsymbol{\theta}) = \frac{1}{2} \sum_{i=1}^n \left( f(\boldsymbol{\theta}, \boldsymbol{x}_i) - \tilde{y}_i \right)^2$. To prevent over-fitting, we suggest the following simple regularization methods that slightly modify this objective:

- **Method 1: Regularization using Distance to Initialization (RDI).** We let the initial parameters $\boldsymbol{\theta}(0)$ be randomly generated, and minimize the following regularized objective:

$$L_\lambda^{\mathsf{RDI}}(\boldsymbol{\theta}) = \frac{1}{2} \sum_{i=1}^n (f(\boldsymbol{\theta}, \boldsymbol{x}_i) - \tilde{y}_i)^2 + \frac{\lambda^2}{2} \| \boldsymbol{\theta} - \boldsymbol{\theta}(0) \|^2. \tag{3}$$

- **Method 2: adding an AUXiliary variable for each training example (AUX).** We add an auxiliary trainable parameter $b_i \in \mathbb{R}$ for each $i \in [n]$, and minimize the following objective:

$$L_\lambda^{\mathsf{AUX}}(\boldsymbol{\theta}, \boldsymbol{b}) = \frac{1}{2} \sum_{i=1}^n (f(\boldsymbol{\theta}, \boldsymbol{x}_i) + \lambda b_i - \tilde{y}_i)^2, \tag{4}$$

  where $\boldsymbol{b} = (b_1, \ldots, b_n)^\top \in \mathbb{R}^n$ is initialized to be $\boldsymbol{0}$.

**Equivalence to kernel ridge regression in wide neural nets.** Now we assume that we are in the NTK regime described in Section 3.2, where the neural net architecture is sufficiently wide so that the first-order approximation (1) is accurate during gradient descent: $f(\boldsymbol{\theta}, \boldsymbol{x}) \approx f(\boldsymbol{\theta}(0), \boldsymbol{x}) + \phi(\boldsymbol{x})^\top(\boldsymbol{\theta} - \boldsymbol{\theta}(0))$. Recall that we have $\phi(\boldsymbol{x}) = \nabla_{\boldsymbol{\theta}} f(\boldsymbol{\theta}(0), \boldsymbol{x})$ which induces the NTK $k(\boldsymbol{x}, \boldsymbol{x}') = \langle \phi(\boldsymbol{x}), \phi(\boldsymbol{x}') \rangle$. Also recall that we can assume near-zero initial output: $f(\boldsymbol{\theta}(0), \boldsymbol{x}) \approx 0$ (see Footnote 2). Therefore we have the approximation:

$$f(\boldsymbol{\theta}, \boldsymbol{x}) \approx \phi(\boldsymbol{x})^\top(\boldsymbol{\theta} - \boldsymbol{\theta}(0)). \tag{5}$$

---

[2]We can ensure small or even zero output at initialization by either multiplying a small factor (Arora et al., 2019a;b), or using the following "difference trick": define the network to be the difference between two networks with the same architecture, i.e., $f(\boldsymbol{\theta}, \boldsymbol{x}) = \frac{\sqrt{2}}{2} g(\boldsymbol{\theta}_1, \boldsymbol{x}) - \frac{\sqrt{2}}{2} g(\boldsymbol{\theta}_2, \boldsymbol{x})$; then initialize $\boldsymbol{\theta}_1$ and $\boldsymbol{\theta}_2$ to be the same (and still random); this ensures $f(\boldsymbol{\theta}(0), \boldsymbol{x}) = 0 \; (\forall \boldsymbol{x})$ at initialization, while keeping the same value of $\langle \phi(\boldsymbol{x}), \phi(\boldsymbol{x}') \rangle$ for both $f$ and $g$. See more details in Appendix A.

[3]For the theoretical results in this paper we use $\ell_2$ loss. In Section 6 we will present experimental results of both $\ell_2$ loss and cross-entropy loss for classification.

Under the approximation (5), it suffices to consider gradient descent on the objectives (3) and (4) using the linearized model instead:

$$\tilde{L}^{\mathsf{RDI}}_\lambda(\boldsymbol{\theta}) = \frac{1}{2} \sum_{i=1}^{n} \left( \boldsymbol{\phi}(\boldsymbol{x}_i)^\top (\boldsymbol{\theta} - \boldsymbol{\theta}(0)) - \tilde{y}_i \right)^2 + \frac{\lambda^2}{2} \|\boldsymbol{\theta} - \boldsymbol{\theta}(0)\|^2 \,,$$

$$\tilde{L}^{\mathsf{AUX}}_\lambda(\boldsymbol{\theta}, \boldsymbol{b}) = \frac{1}{2} \sum_{i=1}^{n} \left( \boldsymbol{\phi}(\boldsymbol{x}_i)^\top (\boldsymbol{\theta} - \boldsymbol{\theta}(0)) + \lambda b_i - \tilde{y}_i \right)^2 \,.$$

The following theorem shows that in either case, gradient descent leads to the same dynamics and converges to the *kernel ridge regression* solution using the NTK.

**Theorem 4.1.** *Fix a learning rate $\eta > 0$. Consider gradient descent on $\tilde{L}^{\mathsf{RDI}}_\lambda$ with initialization $\boldsymbol{\theta}(0)$:*

$$\boldsymbol{\theta}(t+1) = \boldsymbol{\theta}(t) - \eta \nabla_{\boldsymbol{\theta}} \tilde{L}^{\mathsf{RDI}}_\lambda(\boldsymbol{\theta}(t)), \quad t = 0, 1, 2, \dots \tag{6}$$

*and gradient descent on $\tilde{L}^{\mathsf{AUX}}_\lambda(\boldsymbol{\theta}, \boldsymbol{b})$ with initialization $\boldsymbol{\theta}(0)$ and $\boldsymbol{b}(0) = \mathbf{0}$:*

$$\bar{\boldsymbol{\theta}}(0) = \boldsymbol{\theta}(0), \quad \bar{\boldsymbol{\theta}}(t+1) = \bar{\boldsymbol{\theta}}(t) - \eta \nabla_{\boldsymbol{\theta}} \tilde{L}^{\mathsf{AUX}}_\lambda(\bar{\boldsymbol{\theta}}(t), \boldsymbol{b}(t)), \quad t = 0, 1, 2, \dots$$
$$\boldsymbol{b}(0) = \mathbf{0}, \quad \boldsymbol{b}(t+1) = \boldsymbol{b}(t) - \eta \nabla_{\boldsymbol{b}} \tilde{L}^{\mathsf{AUX}}_\lambda(\bar{\boldsymbol{\theta}}(t), \boldsymbol{b}(t)), \quad t = 0, 1, 2, \dots \tag{7}$$

*Then we must have $\boldsymbol{\theta}(t) = \bar{\boldsymbol{\theta}}(t)$ for all $t$. Furthermore, if the learning rate satisfies $\eta \leqslant \frac{1}{\|k(\boldsymbol{X}, \boldsymbol{X})\| + \lambda^2}$, then $\{\boldsymbol{\theta}(t)\}$ converges linearly to a limit solution $\boldsymbol{\theta}^*$ such that:*

$$\boldsymbol{\phi}(\boldsymbol{x})^\top (\boldsymbol{\theta}^* - \boldsymbol{\theta}(0)) = k(\boldsymbol{x}, \boldsymbol{X})^\top \left( k(\boldsymbol{X}, \boldsymbol{X}) + \lambda^2 \boldsymbol{I} \right)^{-1} \tilde{\boldsymbol{y}}, \quad \forall \boldsymbol{x},$$

*where $\tilde{\boldsymbol{y}} = (\tilde{y}_1, \dots, \tilde{y}_n)^\top \in \mathbb{R}^n$.*

*Proof sketch.* The proof is given in Appendix B. A key step is to observe $\bar{\boldsymbol{\theta}}(t) = \boldsymbol{\theta}(0) + \sum_{i=1}^{n} \frac{1}{\lambda} b_i(t) \cdot \boldsymbol{\phi}(\boldsymbol{x}_i)$, from which we can show that $\{\boldsymbol{\theta}(t)\}$ and $\{\bar{\boldsymbol{\theta}}(t)\}$ follow the same update rule. □

Theorem 4.1 indicates that gradient descent on the regularized objectives (3) and (4) both learn approximately the following function at the end of training when the neural net is sufficiently wide:

$$f^*(\boldsymbol{x}) = k(\boldsymbol{x}, \boldsymbol{X})^\top \left( k(\boldsymbol{X}, \boldsymbol{X}) + \lambda^2 \boldsymbol{I} \right)^{-1} \tilde{\boldsymbol{y}}. \tag{8}$$

If no regularization were used, the labels $\tilde{\boldsymbol{y}}$ would be fitted perfectly and the learned function would be $k(\boldsymbol{x}, \boldsymbol{X})^\top \left( k(\boldsymbol{X}, \boldsymbol{X}) \right)^{-1} \tilde{\boldsymbol{y}}$ (c.f. (2)). Therefore the effect of regularization is to add $\lambda^2 \boldsymbol{I}$ to the kernel matrix, and (8) is known as the solution to *kernel ridge regression* in kernel literature. In Section 5, we give a generalization bound of this solution on the clean data distribution, which is comparable to the bound one can obtain even when clean labels are used in training.

**Extension to multiple outputs.** Suppose that the training dataset is $\{(\boldsymbol{x}_i, \tilde{\boldsymbol{y}}_i)\}_{i=1}^{n} \subset \mathbb{R}^d \times \mathbb{R}^K$, and the neural net $\boldsymbol{f}(\boldsymbol{\theta}, \boldsymbol{x})$ has $K$ outputs. On top of the vanilla loss $L(\boldsymbol{\theta}) = \frac{1}{2} \sum_{i=1}^{n} \|\boldsymbol{f}(\boldsymbol{\theta}, \boldsymbol{x}_i) - \tilde{\boldsymbol{y}}_i\|^2$, the two regularization methods RDI and AUX give the following objectives similar to (3) and (4):

$$L^{\mathsf{RDI}}_\lambda(\boldsymbol{\theta}) = \frac{1}{2} \sum_{i=1}^{n} \|\boldsymbol{f}(\boldsymbol{\theta}, \boldsymbol{x}_i) - \tilde{\boldsymbol{y}}_i\|^2 + \frac{\lambda^2}{2} \|\boldsymbol{\theta} - \boldsymbol{\theta}(0)\|^2 \,,$$

$$L^{\mathsf{AUX}}_\lambda(\boldsymbol{\theta}, \boldsymbol{B}) = \frac{1}{2} \sum_{i=1}^{n} \|\boldsymbol{f}(\boldsymbol{\theta}, \boldsymbol{x}_i) + \lambda \boldsymbol{b}_i - \tilde{\boldsymbol{y}}_i\|^2 \,, \quad \boldsymbol{B} = (\boldsymbol{b}_1, \dots, \boldsymbol{b}_n) \in \mathbb{R}^{K \times n} \,.$$

In the NTK regime, both methods lead to the kernel ridge regression solution at each output. Namely, letting $\tilde{\boldsymbol{Y}} = (\tilde{\boldsymbol{y}}_1, \dots, \tilde{\boldsymbol{y}}_n) \in \mathbb{R}^{K \times n}$ be the training target matrix and $\tilde{\boldsymbol{y}}^{(h)} \in \mathbb{R}^n$ be the $h$-th row of $\tilde{\boldsymbol{Y}}$, at the end of training the $h$-th output of the network learns the following function:

$$f^{(h)}(\boldsymbol{x}) = k(\boldsymbol{x}, \boldsymbol{X})^\top \left( k(\boldsymbol{X}, \boldsymbol{X}) + \lambda^2 \boldsymbol{I} \right)^{-1} \tilde{\boldsymbol{y}}^{(h)}, \quad h \in [K]. \tag{9}$$

## 5 GENERALIZATION GUARANTEE ON CLEAN DATA DISTRIBUTION

We show that gradient descent training on noisily labeled data with our regularization methods RDI or AUX leads to a generalization guarantee on the *clean* data distribution. As in Section 4, we consider the NTK regime and let $k(\cdot, \cdot)$ be the NTK corresponding to the neural net. It suffices to analyze the kernel ridge regression predictor, i.e., (8) for single output and (9) for multiple outputs.

We start with a regression setting where labels are real numbers and the noisy label is the true label plus an additive noise (Theorem 5.1). Built on this result, we then provide generalization bounds for the classification settings described in Section 3.1. Omitted proofs in this section are in Appendix C.

**Theorem 5.1** (Additive label noise). *Let $\mathcal{D}$ be a distribution over $\mathbb{R}^d \times [-1, 1]$. Consider the following data generation process: (i) draw $(\boldsymbol{x}, y) \sim \mathcal{D}$, (ii) conditioned on $(\boldsymbol{x}, y)$, let $\varepsilon$ be drawn from a noise distribution $\mathcal{E}_{\boldsymbol{x}, y}$ over $\mathbb{R}$ that may depend on $\boldsymbol{x}$ and $y$, and (iii) let $\tilde{y} = y + \varepsilon$. Suppose that $\mathcal{E}_{\boldsymbol{x}, y}$ has mean $0$ and is subgaussian with parameter $\sigma > 0$, for any $(\boldsymbol{x}, y)$.*

*Let $\{(\boldsymbol{x}_i, y_i, \tilde{y}_i)\}_{i=1}^n$ be i.i.d. samples from the above process. Denote $\boldsymbol{X} = (\boldsymbol{x}_1, \ldots, \boldsymbol{x}_n)$, $\boldsymbol{y} = (y_1, \ldots, y_n)^\top$ and $\tilde{\boldsymbol{y}} = (\tilde{y}_1, \ldots, \tilde{y}_n)^\top$. Consider the kernel ridge regression solution in (8): $f^*(\boldsymbol{x}) = k(\boldsymbol{x}, \boldsymbol{X})^\top \left( k(\boldsymbol{X}, \boldsymbol{X}) + \lambda^2 \boldsymbol{I} \right)^{-1} \tilde{\boldsymbol{y}}$. Suppose that the kernel matrix satisfies $\mathrm{tr}[k(\boldsymbol{X}, \boldsymbol{X})] = O(n)$. Then for any loss function $\ell : \mathbb{R} \times \mathbb{R} \to [0, 1]$ that is 1-Lipschitz in the first argument such that $\ell(y, y) = 0$, with probability at least $1 - \delta$ we have*

$$\mathbb{E}_{(\boldsymbol{x}, y) \sim \mathcal{D}} \left[ \ell(f^*(\boldsymbol{x}), y) \right] \leqslant \frac{\lambda + O(1)}{2} \sqrt{\frac{\boldsymbol{y}^\top (k(\boldsymbol{X}, \boldsymbol{X}))^{-1} \boldsymbol{y}}{n}} + O\left(\frac{\sigma}{\lambda}\right) + \Delta, \tag{10}$$

*where $\Delta = O\left( \sigma \sqrt{\frac{\log(1/\delta)}{n}} + \frac{\sigma}{\lambda} \sqrt{\frac{\log(1/\delta)}{n}} + \sqrt{\frac{\log \frac{n}{\delta \lambda}}{n}} \right)$.*

**Remark 5.1.** *As the number of samples $n \to \infty$, we have $\Delta \to 0$. In order for the second term $O(\frac{\sigma}{\lambda})$ in (10) to go to $0$, we need to choose $\lambda$ to grow with $n$, e.g., $\lambda = n^c$ for some small constant $c > 0$. Then, the only remaining term in (10) to worry about is $\frac{\lambda}{2} \sqrt{\frac{\boldsymbol{y}^\top (k(\boldsymbol{X}, \boldsymbol{X}))^{-1} \boldsymbol{y}}{n}}$. Notice that it depends on the (unobserved) clean labels $\boldsymbol{y}$, instead of the noisy labels $\tilde{\boldsymbol{y}}$. By a very similar proof, one can show that training on the clean labels $\boldsymbol{y}$ (without regularization) leads to a population loss bound $O\left( \sqrt{\frac{\boldsymbol{y}^\top (k(\boldsymbol{X}, \boldsymbol{X}))^{-1} \boldsymbol{y}}{n}} \right)$. In comparison, we can see that even when there is label noise, we only lose a factor of $O(\lambda)$ in the population loss on the clean distribution, which can be chosen as any slow-growing function of $n$. If $\boldsymbol{y}^\top (k(\boldsymbol{X}, \boldsymbol{X}))^{-1} \boldsymbol{y}$ grows much slower than $n$, by choosing an appropriate $\lambda$, our result indicates that the underlying distribution is learnable in presence of additive label noise. See Remark 5.2 for an example.*

**Remark 5.2.** *Arora et al. (2019b) proved that two-layer ReLU neural nets trained with gradient descent can learn a class of smooth functions on the unit sphere. Their proof is by showing $\boldsymbol{y}^\top (k(\boldsymbol{X}, \boldsymbol{X}))^{-1} \boldsymbol{y} = O(1)$ if $y_i = g(\boldsymbol{x}_i)$ $(\forall i \in [n])$ for certain function $g$, where $k(\cdot, \cdot)$ is the NTK corresponding to two-layer ReLU nets. Combined with their result, Theorem 5.1 implies that the same class of functions can be learned by the same network even if the labels are noisy.*

Next we use Theorem 5.1 to provide generalization bounds for the classification settings described in Section 3.1. For binary classification, we treat the labels as $\pm 1$ and consider a single-output neural net; for $K$-class classification, we treat the labels as their one-hot encodings (which are $K$ standard unit vectors in $\mathbb{R}^K$) and consider a $K$-output neural net. Again, we use $\ell_2$ loss and wide neural nets so that it suffices to consider the kernel ridge regression solution ((8) or (9)).

**Theorem 5.2** (Binary classification). *Consider the binary classification setting stated in Section 3.1. Let $\{(\boldsymbol{x}_i, y_i, \tilde{y}_i)\}_{i=1}^n \subset \mathbb{R}^d \times \{\pm 1\} \times \{\pm 1\}$ be i.i.d. samples from that process. Recall that $\Pr[\tilde{y}_i \neq y_i | y_i] = p$ $(0 \leqslant p < \frac{1}{2})$. Denote $\boldsymbol{X} = (\boldsymbol{x}_1, \ldots, \boldsymbol{x}_n)$, $\boldsymbol{y} = (y_1, \ldots, y_n)^\top$ and $\tilde{\boldsymbol{y}} = (\tilde{y}_1, \ldots, \tilde{y}_n)^\top$. Consider the kernel ridge regression solution in (8): $f^*(\boldsymbol{x}) = k(\boldsymbol{x}, \boldsymbol{X})^\top \left( k(\boldsymbol{X}, \boldsymbol{X}) + \lambda^2 \boldsymbol{I} \right)^{-1} \tilde{\boldsymbol{y}}$. Suppose that the kernel matrix satisfies $\mathrm{tr}[k(\boldsymbol{X}, \boldsymbol{X})] = O(n)$. Then with probability at least $1 - \delta$, the classification error of $f^*$ on the clean distribution $\mathcal{D}$ satisfies*

$$\Pr_{(\boldsymbol{x}, y) \sim \mathcal{D}}[\mathrm{sgn}(f^*(\boldsymbol{x})) \neq y] \leqslant \frac{\lambda + O(1)}{2} \sqrt{\frac{\boldsymbol{y}^\top (k(\boldsymbol{X}, \boldsymbol{X}))^{-1} \boldsymbol{y}}{n}} + \frac{1}{1 - 2p} O\left( \frac{\sqrt{p}}{\lambda} + \sqrt{\frac{p \log \frac{1}{\delta}}{n}} + \sqrt{\frac{\log \frac{n}{\delta \lambda}}{n}} \right).$$

**Theorem 5.3** (Multi-class classification). *Consider the $K$-class classification setting stated in Section 3.1. Let $\{(\boldsymbol{x}_i, c_i, \tilde{c}_i)\}_{i=1}^n \subset \mathbb{R}^d \times [K] \times [K]$ be i.i.d. samples from that process. Recall that $\Pr[\tilde{c}_i = c' | c_i = c] = p_{c', c}$ $(\forall c, c' \in [K])$, where the transition probabilities form a matrix $\boldsymbol{P} \in \mathbb{R}^{K \times K}$. Let $\mathsf{gap} = \min_{c, c' \in [K], c \neq c'}(p_{c, c} - p_{c', c})$.*

*Let $\boldsymbol{X} = (\boldsymbol{x}_1, \ldots, \boldsymbol{x}_n)$, and let $\boldsymbol{y}_i = \boldsymbol{e}^{(c_i)} \in \mathbb{R}^K$, $\tilde{\boldsymbol{y}}_i = \boldsymbol{e}^{(\tilde{c}_i)} \in \mathbb{R}^K$ be one-hot label encodings. Denote $\boldsymbol{Y} = (\boldsymbol{y}_1, \ldots, \boldsymbol{y}_n) \in \mathbb{R}^{K \times n}$, $\tilde{\boldsymbol{Y}} = (\tilde{\boldsymbol{y}}_1, \ldots, \tilde{\boldsymbol{y}}_n) \in \mathbb{R}^{K \times n}$, and let $\tilde{\boldsymbol{y}}^{(h)} \in \mathbb{R}^n$ be the $h$-th row of $\tilde{\boldsymbol{Y}}$. Define a matrix $\boldsymbol{Q} = \boldsymbol{P} \cdot \boldsymbol{Y} \in \mathbb{R}^{K \times n}$, and let $\boldsymbol{q}^{(h)} \in \mathbb{R}^n$ be the $h$-th row of $\boldsymbol{Q}$.*

*Consider the kernel ridge regression solution in (9): $f^{(h)}(\boldsymbol{x}) = k(\boldsymbol{x}, \boldsymbol{X})^\top \left( k(\boldsymbol{X}, \boldsymbol{X}) + \lambda^2 \boldsymbol{I} \right)^{-1} \tilde{\boldsymbol{y}}^{(h)}$. Suppose that the kernel matrix satisfies $\mathrm{tr}[k(\boldsymbol{X}, \boldsymbol{X})] = O(n)$. Then with probability at least $1 - \delta$, the classification error of $\boldsymbol{f} = (f^{(h)})_{k=1}^K$ on the clean data distribution $\mathcal{D}$ is bounded as*

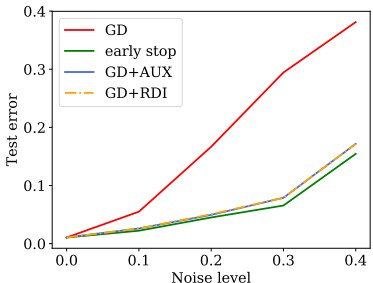
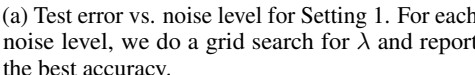

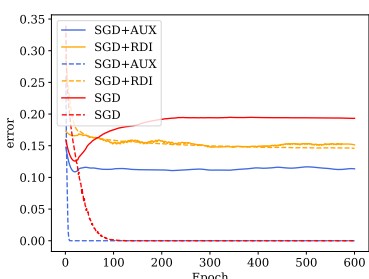

(a) Test error vs. noise level for Setting 1. For each noise level, we do a grid search for $\lambda$ and report the best accuracy.

(b) Training (dashed) & test (solid) errors vs. epoch for Setting 2. Noise rate $= 20\%$, $\lambda = 4$. Training error of `AUX` is measured with auxiliary variables.

Figure 1: Performance on binary classification using $\ell_2$ loss. Setting 1: MNIST; Setting 2: CIFAR.

$$\Pr_{(\boldsymbol{x},c)\sim\mathcal{D}}\left[c \notin \mathrm{argmax}_{h\in[K]}f^{(h)}(\boldsymbol{x})\right]$$

$$\leqslant \frac{1}{\mathsf{gap}}\left(\frac{\lambda+O(1)}{2}\sum_{h=1}^{K}\sqrt{\frac{(\boldsymbol{q}^{(h)})^{\top}(k(\boldsymbol{X},\boldsymbol{X}))^{-1}\boldsymbol{q}^{(h)}}{n}} + K\cdot O\left(\frac{1}{\lambda}+\sqrt{\frac{\log\frac{1}{\delta'}}{n}}+\sqrt{\frac{\log\frac{n}{\delta'\lambda}}{n}}\right)\right).$$

Note that the bounds in Theorems 5.2 and 5.3 only depend on the clean labels instead of the noisy labels, similar to Theorem 5.1.

## 6 EXPERIMENTS

In this section, we empirically verify the effectiveness of our regularization methods `RDI` and `AUX`, and compare them against gradient descent or stochastic gradient descent (GD/SGD) with or without early stopping. We experiment with three settings of increasing complexities:

Setting 1: Binary classification on MNIST ("5" vs. "8") using a two-layer wide fully-connected net.

Setting 2: Binary classification on CIFAR ("airplanes" vs. "automobiles") using a 11-layer convolutional neural net (CNN).

Setting 3: CIFAR-10 classification (10 classes) using standard ResNet-34.

For detailed description see Appendix D. We obtain noisy labels by randomly corrupting correct labels, where noise rate/level is the fraction of corrupted labels (for CIFAR-10, a corrupted label is chosen uniformly from the other 9 classes.)

### 6.1 PERFORMANCE OF REGULARIZATION METHODS

For Setting 1 (binary MNIST), we plot the test errors of different methods under different noise rates in Figure 1a. We observe that both methods `GD+AUX` and `GD+RDI` consistently achieve much lower test error than vanilla GD which over-fits the noisy dataset, and they achieve similar test error to GD with early stopping. We see that `GD+AUX` and `GD+RDI` have essentially the same performance, which verifies our theory of their equivalence in wide networks (Theorem 4.1).

For Setting 2 (binary CIFAR), Figure 1b shows the learning curves (training and test errors) of `SGD`, `SGD+AUX` and `SGD+RDI` for noise rate 20% and $\lambda = 4$. Additional figures for other choices of $\lambda$ are in Figure 5. We again observe that both `SGD+AUX` and `SGD+RDI` outperform vanilla SGD and are comparable to SGD with early stopping. We also observe a discrepancy between `SGD+AUX` and `SGD+RDI`, possibly due to the noise in SGD or the finite width.

Finally, for Setting 3 (CIFAR-10), Table 1 shows the test accuracies of training with and without `AUX`. We train with both mean square error (MSE/$\ell_2$ loss) and categorical cross entropy (CCE) loss. For normal training without `AUX`, we report the test accuracy at the epoch where validation accuracy is maximum (early stopping). For training with `AUX`, we report the test accuracy at the last epoch as well as the best epoch. Figure 2 shows the training curves for noise rate 0.4. We observe that training with `AUX` achieves very good test accuracy – even better than the best accuracy of normal training with early stopping, and better than the recent method of Zhang and Sabuncu (2018) using the same

| Noise rate | 0 | 0.2 | 0.4 | 0.6 |
|---|---|---|---|---|
| Normal CCE (early stop) | 94.05±0.07 | 89.73±0.43 | 86.35±0.47 | 79.13±0.41 |
| Normal MSE (early stop) | 93.88±0.37 | 89.96±0.13 | 85.92±0.32 | 78.68±0.56 |
| CCE+AUX (last) | 94.22±0.10 | 92.07±0.10 | 87.81±0.37 | 82.60±0.29 |
| CCE+AUX (best) | 94.30±0.09 | 92.16±0.08 | 88.61±0.14 | 82.91±0.22 |
| MSE+AUX (last) | 94.25±0.10 | 92.31±0.18 | 88.92±0.30 | 83.90±0.30 |
| MSE+AUX (best) | 94.32±0.06 | 92.40±0.18 | 88.95±0.31 | 83.95±0.30 |
| (Zhang and Sabuncu, 2018) | - | 89.83±0.20 | 87.62±0.26 | 82.70±0.23 |

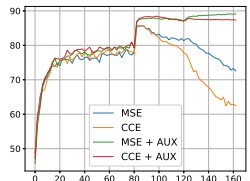

Table 1: CIFAR-10 test accuracies of different methods under different noise rates.

Figure 2: Test accuracy during CIFAR-10 training (noise 0.4).

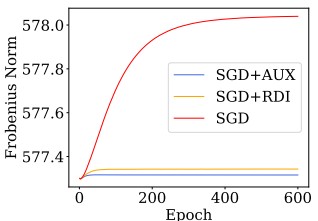 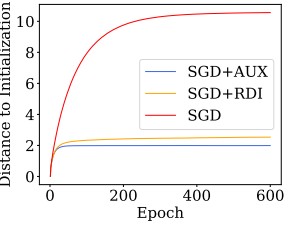

Figure 3: Setting 2, $\|\boldsymbol{W}^{(4)}\|_F$ and $\|\boldsymbol{W}^{(4)} - \boldsymbol{W}^{(4)}(0)\|_F$ during training. Noise = 20%, $\lambda = 4$.

architecture (ResNet-34). Furthermore, AUX does not over-fit (the last epoch performs similarly to the best epoch). In addition, we find that in this setting classification performance is insensitive of whether MSE or CCE is used as the loss function.

## 6.2 DISTANCE OF WEIGHTS TO INITIALIZATION, VERIFICATION OF THE NTK REGIME

We also track how much the weights move during training as a way to see whether the neural net is in the NTK regime. For Settings 1 and 2, we find that the neural nets are likely in or close to the NTK regime because the weight movements are small during training. Figure 3 shows in Setting 2 how much the 4-th layer weights move during training. Additional figures are provided as Figures 6 to 8.

Table 2 summarizes the relationship between the distance to initialization and other hyper-parameters that we observe from various experiments. Note that the weights tend to move more with larger noise level, and AUX and RDI can reduce the moving distance as expected (as shown in Figure 3).

The ResNet-34 in Setting 3 is likely not operating in the NTK regime, so its effectiveness cannot yet be explained by our theory. This is an intriguing direction of future work.

| | # samples | noise level | width | regularization strength $\lambda$ | learning rate |
|---|---|---|---|---|---|
| Distance | ↗ | ↗ | — | ↘ | — |

Table 2: Relationship between distance to initialization at convergence and other hyper-parameters. "↗": positive correlation; "↘": negative correlation; '—': no correlation as long as width is sufficiently large and learning rate is sufficiently small.

## 7 CONCLUSION

Towards understanding generalization of deep neural networks in presence of noisy labels, this paper presents two simple regularization methods and shows that they are theoretically and empirically effective. The theoretical analysis relies on the correspondence between neural networks and NTKs. We believe that a better understanding of such correspondence could help the design of other principled methods in practice. We also observe that our methods can be effective outside the NTK regime. Explaining this theoretically is left for future work.

### ACKNOWLEDGMENTS

This work is supported by NSF, ONR, Simons Foundation, Schmidt Foundation, Mozilla Research, Amazon Research, DARPA and SRC. The authors thank Sanjeev Arora for helpful discussions and suggestions. The authors thank Amazon Web Services for cloud computing time.

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

## A    DIFFERENCE TRICK

We give a quick analysis of the "difference trick" described in Footnote 2, i.e., let $f(\boldsymbol{\theta}, \boldsymbol{x}) = \frac{\sqrt{2}}{2}g(\boldsymbol{\theta}_1, \boldsymbol{x}) - \frac{\sqrt{2}}{2}g(\boldsymbol{\theta}_2, \boldsymbol{x})$ where $\boldsymbol{\theta} = (\boldsymbol{\theta}_1, \boldsymbol{\theta}_2)$, and initialize $\boldsymbol{\theta}_1$ and $\boldsymbol{\theta}_2$ to be the same (and still random). The following lemma implies that the NTKs for $f$ and $g$ are the same.

**Lemma A.1.** *If $\boldsymbol{\theta}_1(0) = \boldsymbol{\theta}_2(0)$, then*

*(i) $f(\boldsymbol{\theta}(0), \boldsymbol{x}) = 0, \ \forall \boldsymbol{x}$;*

*(ii) $\langle \nabla_{\boldsymbol{\theta}} f(\boldsymbol{\theta}(0), \boldsymbol{x}), \nabla_{\boldsymbol{\theta}} f(\boldsymbol{\theta}(0), \boldsymbol{x}') \rangle = \langle \nabla_{\boldsymbol{\theta}_1} g(\boldsymbol{\theta}_1(0), \boldsymbol{x}), \nabla_{\boldsymbol{\theta}_1} g(\boldsymbol{\theta}_1(0), \boldsymbol{x}') \rangle, \ \forall \boldsymbol{x}, \boldsymbol{x}'$.*

*Proof.* (i) holds by definition. For (ii), we can calculate that

$$
\begin{aligned}
&\left\langle \nabla_{\boldsymbol{\theta}} f(\boldsymbol{\theta}(0), \boldsymbol{x}), \nabla_{\boldsymbol{\theta}} f(\boldsymbol{\theta}(0), \boldsymbol{x}') \right\rangle \\
&= \left\langle \nabla_{\boldsymbol{\theta}_1} f(\boldsymbol{\theta}(0), \boldsymbol{x}), \nabla_{\boldsymbol{\theta}_1} f(\boldsymbol{\theta}(0), \boldsymbol{x}') \right\rangle + \left\langle \nabla_{\boldsymbol{\theta}_2} f(\boldsymbol{\theta}(0), \boldsymbol{x}), \nabla_{\boldsymbol{\theta}_2} f(\boldsymbol{\theta}(0), \boldsymbol{x}') \right\rangle \\
&= \frac{1}{2} \left\langle \nabla_{\boldsymbol{\theta}_1} g(\boldsymbol{\theta}_1(0), \boldsymbol{x}), \nabla_{\boldsymbol{\theta}_1} g(\boldsymbol{\theta}_1(0), \boldsymbol{x}) \right\rangle + \frac{1}{2} \left\langle \nabla_{\boldsymbol{\theta}_2} g(\boldsymbol{\theta}_2(0), \boldsymbol{x}), \nabla_{\boldsymbol{\theta}_2} g(\boldsymbol{\theta}_2(0), \boldsymbol{x}') \right\rangle \\
&= \left\langle \nabla_{\boldsymbol{\theta}_1} g(\boldsymbol{\theta}_1(0), \boldsymbol{x}), \nabla_{\boldsymbol{\theta}_1} g(\boldsymbol{\theta}_1(0), \boldsymbol{x}') \right\rangle .
\end{aligned}
$$

$\square$

The above lemma allows us to ensure zero output at initialization while preserving NTK. As a comparison, Chizat and Bach (2018) proposed the following "doubling trick": neurons in the last layer are duplicated, with the new neurons having the same input weights and opposite output weights. This satisfies zero output at initialization, but destroys the NTK. To see why, note that with the "doubling trick", the network will output 0 at initialization no matter what the input to its second to last layer is. Thus the gradients with respect to all parameters that are not in the last two layers are 0.

In our experiments, we observe that the performance of the neural net improves with the "difference trick." See Figure 4. This intuitively makes sense, since the initial network output is independent of the label (only depends on the input) and thus can be viewed as noise. When the width of the neural net is infinity, the initial network output is actually a zero-mean Gaussian process, whose covariance matrix is equal to the NTK contributed by the gradients of parameters in its last layer. Therefore, learning an infinitely wide neural network with nonzero initial output is equivalent to doing kernel regression with an additive correlated Gaussian noise on training and testing labels.

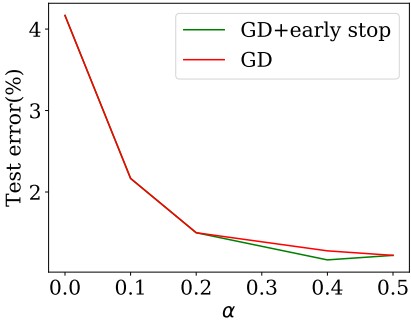

Figure 4: Plot of test error for fully connected two-layer network on MNIST (binary classification between "5" and "8") with difference trick and different mixing coefficients $\alpha$, where $f(\boldsymbol{\theta}_1, \boldsymbol{\theta}_2, \boldsymbol{x}) = \sqrt{\frac{\alpha}{2}}g(\boldsymbol{\theta}_1, \boldsymbol{x}) - \sqrt{\frac{1-\alpha}{2}}g(\boldsymbol{\theta}_1, \boldsymbol{x})$. Note that this parametrization preserves the NTK. The network has 10,000 hidden neurons and we train both layers with gradient descent with fixed learning rate for 5,000 steps. The training loss is less than 0.0001 at the time of stopping. We observe that when $\alpha$ increases, the test error drops because the scale of the initial output of the network goes down.

## B  Missing Proof in Section 4

*Proof of Theorem 4.1.* The gradient of $\tilde{L}_\lambda^{\text{AUX}}(\boldsymbol{\theta}, \boldsymbol{b})$ can be written as

$$\nabla_{\boldsymbol{\theta}} \tilde{L}_\lambda^{\text{AUX}}(\boldsymbol{\theta}, \boldsymbol{b}) = \sum_{i=1}^n a_i \boldsymbol{\phi}(\boldsymbol{x}_i), \quad \nabla_{b_i} \tilde{L}_\lambda^{\text{AUX}}(\boldsymbol{\theta}, \boldsymbol{b}) = \lambda a_i, \quad i = 1, \dots, n,$$

where $a_i = \boldsymbol{\phi}(\boldsymbol{x}_i)^\top (\boldsymbol{\theta} - \boldsymbol{\theta}(0)) + \lambda b_i - \tilde{y}_i \, (i \in [n])$. Therefore we have $\nabla_{\boldsymbol{\theta}} \tilde{L}_\lambda^{\text{AUX}}(\boldsymbol{\theta}, \boldsymbol{b}) = \sum_{i=1}^n \frac{1}{\lambda} \nabla_{b_i} \tilde{L}_\lambda^{\text{AUX}}(\boldsymbol{\theta}, \boldsymbol{b}) \cdot \boldsymbol{\phi}(\boldsymbol{x})$. Then, according to the gradient descent update rule (7), we know that $\bar{\boldsymbol{\theta}}(t)$ and $\boldsymbol{b}(t)$ can always be related by $\bar{\boldsymbol{\theta}}(t) = \boldsymbol{\theta}(0) + \sum_{i=1}^n \frac{1}{\lambda} b_i(t) \cdot \boldsymbol{\phi}(\boldsymbol{x}_i)$. It follows that

$$
\begin{aligned}
\bar{\boldsymbol{\theta}}(t+1) &= \bar{\boldsymbol{\theta}}(t) - \eta \sum_{i=1}^n \left( \boldsymbol{\phi}(\boldsymbol{x}_i)^\top (\bar{\boldsymbol{\theta}}(t) - \boldsymbol{\theta}(0)) + \lambda b_i(t) - \tilde{y}_i \right) \boldsymbol{\phi}(\boldsymbol{x}_i) \\
&= \bar{\boldsymbol{\theta}}(t) - \eta \sum_{i=1}^n \left( \boldsymbol{\phi}(\boldsymbol{x}_i)^\top (\bar{\boldsymbol{\theta}}(t) - \boldsymbol{\theta}(0)) - \tilde{y}_i \right) \boldsymbol{\phi}(\boldsymbol{x}_i) - \eta \lambda \sum_{i=1}^n b_i(t) \boldsymbol{\phi}(\boldsymbol{x}_i) \\
&= \bar{\boldsymbol{\theta}}(t) - \eta \sum_{i=1}^n \left( \boldsymbol{\phi}(\boldsymbol{x}_i)^\top (\bar{\boldsymbol{\theta}}(t) - \boldsymbol{\theta}(0)) - \tilde{y}_i \right) \boldsymbol{\phi}(\boldsymbol{x}_i) - \eta \lambda^2 (\bar{\boldsymbol{\theta}}(t) - \boldsymbol{\theta}(0)).
\end{aligned}
$$

On the other hand, from (6) we have

$$\boldsymbol{\theta}(t+1) = \boldsymbol{\theta}(t) - \eta \sum_{i=1}^n \left( \boldsymbol{\phi}(\boldsymbol{x}_i)^\top (\boldsymbol{\theta}(t) - \boldsymbol{\theta}(0)) - \tilde{y}_i \right) \boldsymbol{\phi}(\boldsymbol{x}_i) - \eta \lambda^2 (\boldsymbol{\theta}(t) - \boldsymbol{\theta}(0)).$$

Comparing the above two equations, we find that $\{\boldsymbol{\theta}(t)\}$ and $\{\bar{\boldsymbol{\theta}}(t)\}$ have the same update rule. Since $\boldsymbol{\theta}(0) = \bar{\boldsymbol{\theta}}(0)$, this proves $\boldsymbol{\theta}(t) = \bar{\boldsymbol{\theta}}(t)$ for all $t$.

Now we prove the second part of the theorem. Notice that $\tilde{L}_\lambda^{\text{RDI}}(\boldsymbol{\theta})$ is a strongly convex quadratic function with Hessian $\nabla_{\boldsymbol{\theta}}^2 \tilde{L}_\lambda^{\text{RDI}}(\boldsymbol{\theta}) = \sum_{i=1}^n \boldsymbol{\phi}(\boldsymbol{x}_i)\boldsymbol{\phi}(\boldsymbol{x}_i)^\top + \lambda^2 \boldsymbol{I} = \boldsymbol{Z}\boldsymbol{Z}^\top + \lambda^2 \boldsymbol{I}$, where $\boldsymbol{Z} = (\boldsymbol{\phi}(\boldsymbol{x}_1), \dots, \boldsymbol{\phi}(\boldsymbol{x}_n))$. From the classical convex optimization theory, as long as $\eta \leqslant \frac{1}{\|\boldsymbol{Z}\boldsymbol{Z}^\top + \lambda^2 \boldsymbol{I}\|} = \frac{1}{\|\boldsymbol{Z}^\top \boldsymbol{Z} + \lambda^2 \boldsymbol{I}\|} = \frac{1}{\|k(\boldsymbol{X}, \boldsymbol{X})\| + \lambda^2}$, gradient descent converges linearly to the unique optimum $\boldsymbol{\theta}^*$ of $\tilde{L}_\lambda^{\text{RDI}}(\boldsymbol{\theta})$, which can be easily obtained:

$$\boldsymbol{\theta}^* = \boldsymbol{\theta}(0) + \boldsymbol{Z}(k(\boldsymbol{X}, \boldsymbol{X}) + \lambda^2 \boldsymbol{I})^{-1} \tilde{\boldsymbol{y}}.$$

Then we have

$$\boldsymbol{\phi}(\boldsymbol{x})^\top (\boldsymbol{\theta}^* - \boldsymbol{\theta}(0)) = \boldsymbol{\phi}(\boldsymbol{x})^\top \boldsymbol{Z}(k(\boldsymbol{X}, \boldsymbol{X}) + \lambda^2 \boldsymbol{I})^{-1} \tilde{\boldsymbol{y}} = k(\boldsymbol{x}, \boldsymbol{X})^\top \left( k(\boldsymbol{X}, \boldsymbol{X}) + \lambda^2 \boldsymbol{I} \right)^{-1} \tilde{\boldsymbol{y}},$$

finishing the proof. $\qquad\square$

## C  Missing Proofs in Section 5

### C.1  Proof of Theorem 5.1

Define $\varepsilon_i = \tilde{y}_i - y_i \, (i \in [n])$, and $\boldsymbol{\varepsilon} = (\varepsilon_1, \dots, \varepsilon_n)^\top = \tilde{\boldsymbol{y}} - \boldsymbol{y}$.

We first prove two lemmas.

**Lemma C.1.** *With probability at least $1 - \delta$, we have*

$$\sqrt{\sum_{i=1}^n \left( f^*(\boldsymbol{x}_i) - y_i \right)^2} \leqslant \frac{\lambda}{2} \sqrt{\boldsymbol{y}^\top (k(\boldsymbol{X}, \boldsymbol{X}))^{-1} \boldsymbol{y}} + \frac{\sigma}{2\lambda} \sqrt{\text{tr}[k(\boldsymbol{X}, \boldsymbol{X})]} + \sigma \sqrt{2 \log(1/\delta)}.$$

*Proof.* In this proof we are conditioned on $\boldsymbol{X}$ and $\boldsymbol{y}$, and only consider the randomness in $\tilde{\boldsymbol{y}}$ given $\boldsymbol{X}$ and $\boldsymbol{y}$.

First of all, we can write

$$(f^*(\boldsymbol{x}_1), \dots, f^*(\boldsymbol{x}_n))^\top = k(\boldsymbol{X}, \boldsymbol{X}) \left( k(\boldsymbol{X}, \boldsymbol{X}) + \lambda^2 \boldsymbol{I} \right)^{-1} \tilde{\boldsymbol{y}},$$

so we can write the training loss on true labels $\boldsymbol{y}$ as

$$
\begin{aligned}
\sqrt{\sum_{i=1}^{n} \left(f^*(\boldsymbol{x}_i) - y_i\right)^2} &= \left\| k(\boldsymbol{X}, \boldsymbol{X}) \left(k(\boldsymbol{X}, \boldsymbol{X}) + \lambda^2 \boldsymbol{I}\right)^{-1} \tilde{\boldsymbol{y}} - \boldsymbol{y} \right\| \\
&= \left\| k(\boldsymbol{X}, \boldsymbol{X}) \left(k(\boldsymbol{X}, \boldsymbol{X}) + \lambda^2 \boldsymbol{I}\right)^{-1} (\boldsymbol{y} + \varepsilon) - \boldsymbol{y} \right\| \\
&= \left\| k(\boldsymbol{X}, \boldsymbol{X}) \left(k(\boldsymbol{X}, \boldsymbol{X}) + \lambda^2 \boldsymbol{I}\right)^{-1} \varepsilon - \lambda^2 \left(k(\boldsymbol{X}, \boldsymbol{X}) + \lambda^2 \boldsymbol{I}\right)^{-1} \boldsymbol{y} \right\| \\
&\leqslant \left\| k(\boldsymbol{X}, \boldsymbol{X}) \left(k(\boldsymbol{X}, \boldsymbol{X}) + \lambda^2 \boldsymbol{I}\right)^{-1} \varepsilon \right\| + \lambda^2 \left\| \left(k(\boldsymbol{X}, \boldsymbol{X}) + \lambda^2 \boldsymbol{I}\right)^{-1} \boldsymbol{y} \right\|.
\end{aligned}
\tag{11}
$$

Next, since $\varepsilon$, conditioned on $\boldsymbol{X}$ and $\boldsymbol{y}$, has independent and subgaussian entries (with parameter $\sigma$), by (Hsu et al., 2012), for any symmetric matrix $\boldsymbol{A}$, with probability at least $1 - \delta$,

$$
\|\boldsymbol{A}\varepsilon\| \leqslant \sigma \sqrt{\operatorname{tr}[\boldsymbol{A}^2] + 2\sqrt{\operatorname{tr}[\boldsymbol{A}^4]\log(1/\delta)} + 2\|\boldsymbol{A}^2\|\log(1/\delta)}.
\tag{12}
$$

Let $\boldsymbol{A} = k(\boldsymbol{X}, \boldsymbol{X}) \left(k(\boldsymbol{X}, \boldsymbol{X}) + \lambda^2 \boldsymbol{I}\right)^{-1}$ and let $\lambda_1, \ldots, \lambda_n > 0$ be the eigenvalues of $k(\boldsymbol{X}, \boldsymbol{X})$. We have

$$
\begin{aligned}
\operatorname{tr}[\boldsymbol{A}^2] &= \sum_{i=1}^{n} \frac{\lambda_i^2}{(\lambda_i + \lambda^2)^2} \leqslant \sum_{i=1}^{n} \frac{\lambda_i^2}{4\lambda_i \cdot \lambda^2} = \frac{\operatorname{tr}[k(\boldsymbol{X}, \boldsymbol{X})]}{4\lambda^2}, \\
\operatorname{tr}[\boldsymbol{A}^4] &= \sum_{i=1}^{n} \frac{\lambda_i^4}{(\lambda_i + \lambda^2)^4} \leqslant \sum_{i=1}^{n} \frac{\lambda_i^4}{4^4 \lambda^2 \left(\frac{\lambda_i}{3}\right)^3} \leqslant \frac{\operatorname{tr}[k(\boldsymbol{X}, \boldsymbol{X})]}{9\lambda^2}, \\
\|\boldsymbol{A}^2\| &\leqslant 1.
\end{aligned}
\tag{13}
$$

Therefore,

$$
\begin{aligned}
\left\| k(\boldsymbol{X}, \boldsymbol{X}) \left(k(\boldsymbol{X}, \boldsymbol{X}) + \lambda^2 \boldsymbol{I}\right)^{-1} \varepsilon \right\| &\leqslant \sigma \sqrt{\frac{\operatorname{tr}[k(\boldsymbol{X}, \boldsymbol{X})]}{4\lambda^2} + 2\sqrt{\frac{\operatorname{tr}[k(\boldsymbol{X}, \boldsymbol{X})]\log(1/\delta)}{9\lambda^2}} + 2\log(1/\delta)} \\
&\leqslant \sigma \left( \sqrt{\frac{\operatorname{tr}[k(\boldsymbol{X}, \boldsymbol{X})]}{4\lambda^2}} + \sqrt{2\log(1/\delta)} \right).
\end{aligned}
$$

Finally, since $\left(k(\boldsymbol{X}, \boldsymbol{X}) + \lambda^2 \boldsymbol{I}\right)^{-2} \preceq \frac{1}{4\lambda^2} \left(k(\boldsymbol{X}, \boldsymbol{X})\right)^{-1}$ (note $(\lambda_i + \lambda^2)^2 \geqslant 4\lambda_i \cdot \lambda^2$), we have

$$
\lambda^2 \left\| \left(k(\boldsymbol{X}, \boldsymbol{X}) + \lambda^2 \boldsymbol{I}\right)^{-1} \boldsymbol{y} \right\| = \lambda^2 \sqrt{\boldsymbol{y}^\top \left(k(\boldsymbol{X}, \boldsymbol{X}) + \lambda^2 \boldsymbol{I}\right)^{-2} \boldsymbol{y}} \leqslant \frac{\lambda}{2} \sqrt{\boldsymbol{y}^\top (k(\boldsymbol{X}, \boldsymbol{X}))^{-1} \boldsymbol{y}}.
\tag{14}
$$

The proof is finished by combining (11), (13) and (14). $\qquad\square$

Let $\mathcal{H}$ be the reproducing kernel Hilbert space (RKHS) corresponding to the kernel $k(\cdot, \cdot)$. Recall that the RKHS norm of a function $f(\boldsymbol{x}) = \boldsymbol{\alpha}^\top k(\boldsymbol{x}, \boldsymbol{X})$ is

$$
\|f\|_{\mathcal{H}} = \sqrt{\boldsymbol{\alpha}^\top k(\boldsymbol{X}, \boldsymbol{X}) \boldsymbol{\alpha}}.
$$

**Lemma C.2.** *With probability at least $1 - \delta$, we have*

$$
\|f^*\|_{\mathcal{H}} \leqslant \sqrt{\boldsymbol{y}^\top \left(k(\boldsymbol{X}, \boldsymbol{X}) + \lambda^2 \boldsymbol{I}\right)^{-1} \boldsymbol{y}} + \frac{\sigma}{\lambda} \left( \sqrt{n} + \sqrt{2\log(1/\delta)} \right)
$$

*Proof.* In this proof we are still conditioned on $\boldsymbol{X}$ and $\boldsymbol{y}$, and only consider the randomness in $\tilde{\boldsymbol{y}}$ given $\boldsymbol{X}$ and $\boldsymbol{y}$. Note that $f^*(\boldsymbol{x}) = \boldsymbol{\alpha}^\top k(\boldsymbol{x}, \boldsymbol{X})$ with $\boldsymbol{\alpha} = \left(k(\boldsymbol{X}, \boldsymbol{X}) + \lambda^2 \boldsymbol{I}\right)^{-1} \tilde{\boldsymbol{y}}$. Since $\left(k(\boldsymbol{X}, \boldsymbol{X}) + \lambda^2 \boldsymbol{I}\right)^{-1} \preceq (k(\boldsymbol{X}, \boldsymbol{X}))^{-1}$ and $\left(k(\boldsymbol{X}, \boldsymbol{X}) + \lambda^2 \boldsymbol{I}\right)^{-1} \preceq \frac{1}{\lambda^2} \boldsymbol{I}$, we can bound

$$
\begin{aligned}
\|f^*\|_{\mathcal{H}} &= \sqrt{\tilde{\boldsymbol{y}}^\top \left(k(\boldsymbol{X}, \boldsymbol{X}) + \lambda^2 \boldsymbol{I}\right)^{-1} k(\boldsymbol{X}, \boldsymbol{X}) \left(k(\boldsymbol{X}, \boldsymbol{X}) + \lambda^2 \boldsymbol{I}\right)^{-1} \tilde{\boldsymbol{y}}} \\
&\leqslant \sqrt{(\boldsymbol{y} + \varepsilon)^\top \left(k(\boldsymbol{X}, \boldsymbol{X}) + \lambda^2 \boldsymbol{I}\right)^{-1} (\boldsymbol{y} + \varepsilon)}
\end{aligned}
$$

$$\leqslant \sqrt{\boldsymbol{y}^\top \left(k(\boldsymbol{X}, \boldsymbol{X}) + \lambda^2 \boldsymbol{I}\right)^{-1} \boldsymbol{y}} + \sqrt{\varepsilon^\top \left(k(\boldsymbol{X}, \boldsymbol{X}) + \lambda^2 \boldsymbol{I}\right)^{-1} \varepsilon}$$

$$\leqslant \sqrt{\boldsymbol{y}^\top \left(k(\boldsymbol{X}, \boldsymbol{X}) + \lambda^2 \boldsymbol{I}\right)^{-1} \boldsymbol{y}} + \frac{\sqrt{\varepsilon^\top \varepsilon}}{\lambda}.$$

Since $\varepsilon$ has independent and subgaussian (with parameter $\sigma$) coordinates, using (12) with $\boldsymbol{A} = \boldsymbol{I}$, with probability at least $1 - \delta$ we have

$$\sqrt{\varepsilon^\top \varepsilon} \leqslant \sigma \left(\sqrt{n} + \sqrt{2 \log(1/\delta)}\right). \qquad \square$$

Now we prove Theorem 5.1.

*Proof of Theorem 5.1.* First, by Lemma C.1, with probability $1 - \delta/3$,

$$\sqrt{\sum_{i=1}^n (f^*(\boldsymbol{x}_i) - y_i)^2} \leqslant \frac{\lambda}{2} \sqrt{\boldsymbol{y}^\top (k(\boldsymbol{X}, \boldsymbol{X}))^{-1} \boldsymbol{y}} + \frac{\sigma}{2\lambda} \sqrt{\mathrm{tr}[k(\boldsymbol{X}, \boldsymbol{X})]} + \sigma \sqrt{2 \log(3/\delta)},$$

which implies that the training error on the true labels under loss function $\ell$ is bounded as

$$\frac{1}{n} \sum_{i=1}^n \ell(f^*(\boldsymbol{x}_i), y_i) = \frac{1}{n} \sum_{i=1}^n \left(\ell(f^*(\boldsymbol{x}_i), y_i) - \ell(y_i, y_i)\right)$$

$$\leqslant \frac{1}{n} \sum_{i=1}^n |f^*(\boldsymbol{x}_i) - y_i|$$

$$\leqslant \frac{1}{\sqrt{n}} \sqrt{\sum_{i=1}^n |f^*(\boldsymbol{x}_i) - y_i|^2}$$

$$\leqslant \frac{\lambda}{2} \sqrt{\frac{\boldsymbol{y}^\top (k(\boldsymbol{X}, \boldsymbol{X}))^{-1} \boldsymbol{y}}{n}} + \frac{\sigma}{2\lambda} \sqrt{\frac{\mathrm{tr}[k(\boldsymbol{X}, \boldsymbol{X})]}{n}} + \sigma \sqrt{\frac{2 \log(3/\delta)}{n}}.$$

Next, for function class $\mathcal{F}_B = \{f(\boldsymbol{x}) = \boldsymbol{\alpha}^\top k(\boldsymbol{x}, \boldsymbol{X}) : \|f\|_{\mathcal{H}} \leqslant B\}$, Bartlett and Mendelson (2002) showed that its empirical Rademacher complexity can be bounded as

$$\hat{\mathcal{R}}_S(\mathcal{F}_B) \triangleq \frac{1}{n} \underset{\boldsymbol{\gamma} \sim \{\pm 1\}^n}{\mathbb{E}} \left[ \sup_{f \in \mathcal{F}_B} \sum_{i=1}^n f(\boldsymbol{x}_i) \gamma_i \right] \leqslant \frac{B \sqrt{\mathrm{tr}[k(\boldsymbol{X}, \boldsymbol{X})]}}{n}.$$

By Lemma C.2, with probability at least $1 - \delta/3$ we have

$$\|f^*\|_{\mathcal{H}} \leqslant \sqrt{\boldsymbol{y}^\top \left(k(\boldsymbol{X}, \boldsymbol{X}) + \lambda^2 \boldsymbol{I}\right)^{-1} \boldsymbol{y}} + \frac{\sigma}{\lambda} \left(\sqrt{n} + \sqrt{2 \log(3/\delta)}\right) \triangleq B'.$$

We also recall the standard generalization bound from Rademacher complexity (see e.g. (Mohri et al., 2012)): with probability at least $1 - \delta$, we have

$$\sup_{f \in \mathcal{F}} \left\{ \mathbb{E}_{(\boldsymbol{x}, y) \sim \mathcal{D}}[\ell(f(\boldsymbol{x}), y)] - \frac{1}{n} \sum_{i=1}^n \ell(f(\boldsymbol{x}_i), y_i) \right\} \leqslant 2 \hat{\mathcal{R}}_S(\mathcal{F}) + 3 \sqrt{\frac{\log(2/\delta)}{2n}}. \qquad (15)$$

Then it is tempting to apply the above bound on the function class $\mathcal{F}_{B'}$ which contains $f^*$. However, we are not yet able to do so, because $B'$ depends on the data $\boldsymbol{X}$ and $\boldsymbol{y}$. To deal with this, we use a standard $\epsilon$-net argument on the interval that $B'$ must lie in: (note that $\|\boldsymbol{y}\| = O(\sqrt{n})$)

$$B' \in \left[ \frac{\sigma}{\lambda} \left(\sqrt{n} + \sqrt{2 \log(3/\delta)}\right), \frac{\sigma}{\lambda} \left(\sqrt{n} + \sqrt{2 \log(3/\delta)}\right) + O\left(\frac{n}{\lambda}\right) \right].$$

The above interval has length $O\left(\frac{n}{\lambda}\right)$, so its $\epsilon$-net $\mathcal{N}$ has size $O\left(\frac{n}{\epsilon \lambda}\right)$. Using a union bound, we apply the generalization bound (15) simultaneously on $\mathcal{F}_B$ for all $B \in \mathcal{N}$: with probability at least $1 - \delta/3$ we have

$$\sup_{f_B \in \mathcal{F}} \left\{ \mathbb{E}_{(\boldsymbol{x}, y) \sim \mathcal{D}}[\ell(f(\boldsymbol{x}), y)] - \frac{1}{n} \sum_{i=1}^n \ell(f(\boldsymbol{x}_i), y_i) \right\} \leqslant 2 \hat{\mathcal{R}}_S(\mathcal{F}_B) + O\left(\sqrt{\frac{\log \frac{n}{\epsilon \delta \lambda}}{n}}\right), \quad \forall B \in \mathcal{N}.$$

By definition there exists $B \in \mathcal{N}$ such that $B' \leqslant B \leqslant B' + \epsilon$. Then we also have $f^* \in \mathcal{F}_B$. Using the above bound on this particular $B$, and putting all parts together, we know that with probability at least $1 - \delta$,

$$\mathbb{E}_{(\boldsymbol{x},y)\sim\mathcal{D}}[\ell(f^*(\boldsymbol{x}),y)]$$

$$\leqslant \frac{1}{n}\sum_{i=1}^{n}\ell(f^*(\boldsymbol{x}_i),y_i) + 2\hat{\mathcal{R}}_S(\mathcal{F}_B) + O\left(\sqrt{\frac{\log\frac{n}{\epsilon\delta\lambda}}{n}}\right)$$

$$\leqslant \frac{\lambda}{2}\sqrt{\frac{\boldsymbol{y}^\top(k(\boldsymbol{X},\boldsymbol{X}))^{-1}\boldsymbol{y}}{n}} + \frac{\sigma}{2\lambda}\sqrt{\frac{\mathrm{tr}[k(\boldsymbol{X},\boldsymbol{X})]}{n}} + \sigma\sqrt{\frac{2\log(3/\delta)}{n}}$$

$$+ \frac{2(B'+\epsilon)\sqrt{\mathrm{tr}[k(\boldsymbol{X},\boldsymbol{X})]}}{n} + O\left(\sqrt{\frac{\log\frac{n}{\epsilon\delta\lambda}}{n}}\right)$$

$$\leqslant \frac{\lambda}{2}\sqrt{\frac{\boldsymbol{y}^\top(k(\boldsymbol{X},\boldsymbol{X}))^{-1}\boldsymbol{y}}{n}} + \frac{\sigma}{2\lambda}\sqrt{\frac{\mathrm{tr}[k(\boldsymbol{X},\boldsymbol{X})]}{n}} + \sigma\sqrt{\frac{2\log(3/\delta)}{n}}$$

$$+ \frac{2\sqrt{\mathrm{tr}[k(\boldsymbol{X},\boldsymbol{X})]}}{n}\left(\sqrt{\boldsymbol{y}^\top(k(\boldsymbol{X},\boldsymbol{X}))^{-1}\boldsymbol{y}} + \frac{\sigma}{\lambda}\left(\sqrt{n} + \sqrt{2\log(3/\delta)}\right) + \epsilon\right) + O\left(\sqrt{\frac{\log\frac{n}{\epsilon\delta\lambda}}{n}}\right)$$

$$= \frac{\lambda}{2}\sqrt{\frac{\boldsymbol{y}^\top(k(\boldsymbol{X},\boldsymbol{X}))^{-1}\boldsymbol{y}}{n}} + \frac{5\sigma}{2\lambda}\sqrt{\frac{\mathrm{tr}[k(\boldsymbol{X},\boldsymbol{X})]}{n}} + \sigma\sqrt{\frac{2\log(3/\delta)}{n}}$$

$$+ \frac{2\sqrt{\mathrm{tr}[k(\boldsymbol{X},\boldsymbol{X})]}}{n}\left(\sqrt{\boldsymbol{y}^\top(k(\boldsymbol{X},\boldsymbol{X}))^{-1}\boldsymbol{y}} + \frac{\sigma}{\lambda}\sqrt{2\log(3/\delta)} + \epsilon\right) + O\left(\sqrt{\frac{\log\frac{n}{\epsilon\delta\lambda}}{n}}\right).$$

Then, using $\mathrm{tr}[k(\boldsymbol{X},\boldsymbol{X})] = O(n)$ and choosing $\epsilon = 1$, we obtain

$$\mathbb{E}_{(\boldsymbol{x},y)\sim\mathcal{D}}[\ell(f^*(\boldsymbol{x}),y)]$$

$$\leqslant \frac{\lambda}{2}\sqrt{\frac{\boldsymbol{y}^\top(k(\boldsymbol{X},\boldsymbol{X}))^{-1}\boldsymbol{y}}{n}} + O\left(\frac{\sigma}{\lambda}\right) + \sigma\sqrt{\frac{2\log(3/\delta)}{n}}$$

$$+ O\left(\sqrt{\frac{\boldsymbol{y}^\top(k(\boldsymbol{X},\boldsymbol{X}))^{-1}\boldsymbol{y}}{n}}\right) + O\left(\frac{\sigma}{\lambda\sqrt{n}}\sqrt{2\log(3/\delta)}\right) + O\left(\frac{1}{\sqrt{n}}\right) + O\left(\sqrt{\frac{\log\frac{n}{\delta\lambda}}{n}}\right)$$

$$\leqslant \frac{\lambda + O(1)}{2}\sqrt{\frac{\boldsymbol{y}^\top(k(\boldsymbol{X},\boldsymbol{X}))^{-1}\boldsymbol{y}}{n}} + O\left(\frac{\sigma}{\lambda} + \sigma\sqrt{\frac{\log(1/\delta)}{n}} + \frac{\sigma}{\lambda}\sqrt{\frac{\log(1/\delta)}{n}} + \sqrt{\frac{\log\frac{n}{\delta\lambda}}{n}}\right).$$

$\square$

## C.2 PROOF OF THEOREM 5.2

*Proof of Theorem 5.2.* We will apply Theorem 5.1 in this proof. Note that we cannot directly apply it on $\{(\boldsymbol{x}_i, y_i, \tilde{y}_i)\}$ because the mean of $\tilde{y}_i - y_i$ is non-zero (conditioned on $(\boldsymbol{x}_i, y_i)$). Nevertheless, this issue can be resolved by considering $\{(\boldsymbol{x}_i, (1-2p)y_i, \tilde{y}_i)\}$ instead.[4] Then we can easily check that conditioned on $y_i$, $\tilde{y}_i - (1-2p)y_i$ has mean 0 and is subgaussian with parameter $\sigma = O(\sqrt{p})$. With this change, we are ready to apply Theorem 5.1.

Define the following ramp loss for $u \in \mathbb{R}, \bar{y} \in \{\pm(1-2p)\}$:

$$\ell^{\mathrm{ramp}}(u,\bar{y}) = \begin{cases} (1-2p), & u\bar{y} \leqslant 0, \\ (1-2p) - \frac{1}{1-2p}u\bar{y}, & 0 < u\bar{y} < (1-2p)^2, \\ 0, & u\bar{y} \geqslant (1-2p)^2. \end{cases}$$

It is easy to see that $\ell^{\mathrm{ramp}}(u,\bar{y})$ is 1-Lipschitz in $u$ for $\bar{y} \in \{\pm(1-2p)\}$, and satisfies $\ell^{\mathrm{ramp}}(\bar{y},\bar{y}) = 0$ for $\bar{y} \in \{\pm(1-2p)\}$. Then by Theorem 5.1, with probability at least $1 - \delta$,

$$\mathbb{E}_{(\boldsymbol{x},y)\sim\mathcal{D}}\left[\ell^{\mathrm{ramp}}(f^*(\boldsymbol{x}),(1-2p)y)\right]$$

---

[4]For binary classification, only the sign of the label matters, so we can imagine that the true labels are from $\{\pm(1-2p)\}$ instead of $\{\pm1\}$, without changing the classification problem.

$$\leqslant \frac{\lambda + O(1)}{2}\sqrt{\frac{(1-2p)\boldsymbol{y}^\top(k(\boldsymbol{X},\boldsymbol{X}))^{-1}\cdot(1-2p)\boldsymbol{y}}{n}} + O\left(\frac{\sqrt{p}}{\lambda} + \sqrt{\frac{p\log(1/\delta)}{n}} + \sqrt{\frac{\log\frac{n}{\delta\lambda}}{n}}\right)$$

$$= \frac{(1-2p)(\lambda + O(1))}{2}\sqrt{\frac{\boldsymbol{y}^\top(k(\boldsymbol{X},\boldsymbol{X}))^{-1}\boldsymbol{y}}{n}} + O\left(\frac{\sqrt{p}}{\lambda} + \sqrt{\frac{p\log(1/\delta)}{n}} + \sqrt{\frac{\log\frac{n}{\delta\lambda}}{n}}\right).$$

Note that $\ell^{\mathrm{ramp}}$ also bounds the 0-1 loss (classification error) as

$$\ell^{\mathrm{ramp}}(u, (1-2p)y) \geqslant (1-2p)\mathbb{I}[\mathrm{sgn}(u) \neq \mathrm{sgn}(y)] = (1-2p)\mathbb{I}[\mathrm{sgn}(u) \neq y], \quad \forall u \in \mathbb{R}, y \in \{\pm 1\}.$$

Then the conclusion follows. $\qquad\square$

## C.3   PROOF OF THEOREM 5.3

*Proof of Theorem 5.3.* By definition, we have $\mathbb{E}[\tilde{\boldsymbol{y}}_i|c_i] = \mathbb{E}[e^{(\tilde{c}_i)}|c_i] = \boldsymbol{p}_{c_i}$ (the $c_i$-th column of $\boldsymbol{P}$). This means $\mathbb{E}[\tilde{\boldsymbol{Y}}|\boldsymbol{Q}] = \boldsymbol{Q}$. Therefore we can view $\boldsymbol{Q}$ as an encoding of clean labels, and then the observed noisy labels $\tilde{\boldsymbol{Y}}$ is $\boldsymbol{Q}$ plus a zero-mean noise. (The noise is always bounded by 1 so is subgaussian with parameter 1.) This enables us to apply Theorem 5.1 to each $f^{(h)}$, which says that with probability at least $1 - \delta'$,

$$\mathbb{E}_{(\boldsymbol{x},c)\sim\mathcal{D}}\left|f^{(h)}(\boldsymbol{x}) - p_{h,c}\right| \leqslant \frac{\lambda + O(1)}{2}\sqrt{\frac{(\boldsymbol{q}^{(h)})^\top(k(\boldsymbol{X},\boldsymbol{X}))^{-1}\boldsymbol{q}^{(h)}}{n}} + O\left(\frac{1}{\lambda} + \sqrt{\frac{\log\frac{1}{\delta'}}{n}} + \sqrt{\frac{\log\frac{n}{\delta'\lambda}}{n}}\right).$$

Letting $\delta' = \frac{\delta}{K}$ and taking a union bound over $h \in [K]$, we know that the above bound holds for every $h$ simultaneously with probability at least $1 - \delta$.

Now we proceed to bound the classification error. Note that $c \notin \mathrm{argmax}_{h\in[K]}f^{(h)}(\boldsymbol{x})$ implies $\sum_{h=1}^K \left|f^{(h)}(\boldsymbol{x}) - p_{h,c}\right| \geqslant \mathsf{gap}$. Therefore the classification error can be bounded as

$$\Pr_{(\boldsymbol{x},c)\sim\mathcal{D}}\left[c \notin \mathrm{argmax}_{h\in[K]}f^{(h)}(\boldsymbol{x})\right] \leqslant \Pr_{(\boldsymbol{x},c)\sim\mathcal{D}}\left[\sum_{h=1}^K \left|f^{(h)}(\boldsymbol{x}) - p_{h,c}\right| \geqslant \mathsf{gap}\right]$$

$$\leqslant \frac{1}{\mathsf{gap}}\mathbb{E}_{(\boldsymbol{x},c)\sim\mathcal{D}}\left[\sum_{h=1}^K \left|f^{(h)}(\boldsymbol{x}) - p_{h,c}\right|\right] = \frac{1}{\mathsf{gap}}\sum_{h=1}^K \mathbb{E}_{(\boldsymbol{x},c)\sim\mathcal{D}}\left[\left|f^{(h)}(\boldsymbol{x}) - p_{h,c}\right|\right]$$

$$\leqslant \frac{1}{\mathsf{gap}}\left(\frac{\lambda + O(1)}{2}\sum_{h=1}^K \sqrt{\frac{(\boldsymbol{q}^{(h)})^\top(k(\boldsymbol{X},\boldsymbol{X}))^{-1}\boldsymbol{q}^{(h)}}{n}} + K \cdot O\left(\frac{1}{\lambda} + \sqrt{\frac{\log\frac{1}{\delta'}}{n}} + \sqrt{\frac{\log\frac{n}{\delta'\lambda}}{n}}\right)\right),$$

completing the proof. $\qquad\square$

## D   EXPERIMENT DETAILS AND ADDITIONAL FIGURES

In Setting 1, we train a two-layer neural network with 10,000 hidden neurons on MNIST ("5" vs "8"). In Setting 2, we train a CNN, which has 192 channels for each of its 11 layers, on CIFAR ("airplanes" vs "automobiles"). We do not have biases in these two networks. In Setting 3, we use the standard ResNet-34.

In Settings 1 and 2, we use a fixed learning rate for GD or SGD, and we do not use tricks like batch normalization, data augmentation, dropout, etc., except the difference trick in Appendix A. We also freeze the first and the last layer of the CNN and the second layer of the fully-connected net.[5]

In Setting 3, we use SGD with 0.9 momentum, weight decay of $5 \times 10^{-4}$, and batch size 128. The learning rate is 0.1 initially and is divided by 10 after 82 and 123 epochs (164 in total). Since we

---

[5]This is because the NTK scaling at initialization balances the norm of the gradient in each layer, and the first and the last layer weights have smaller norm than intermediate layers. Since different weights are expected to move the same amount during training (Du et al., 2018a), the first and last layers move relatively further from their initializations. By freezing them during training, we make the network closer to the NTK regime.

observe little over-fitting to noise in the first stage of training (before learning rate decay), we restrict the regularization power of AUX by applying weight decay on auxiliary variables, and dividing their weight decay factor by 10 after each learning rate decay.

See Figures 5 to 8 for additional results for Setting 2 with different $\lambda$'s.

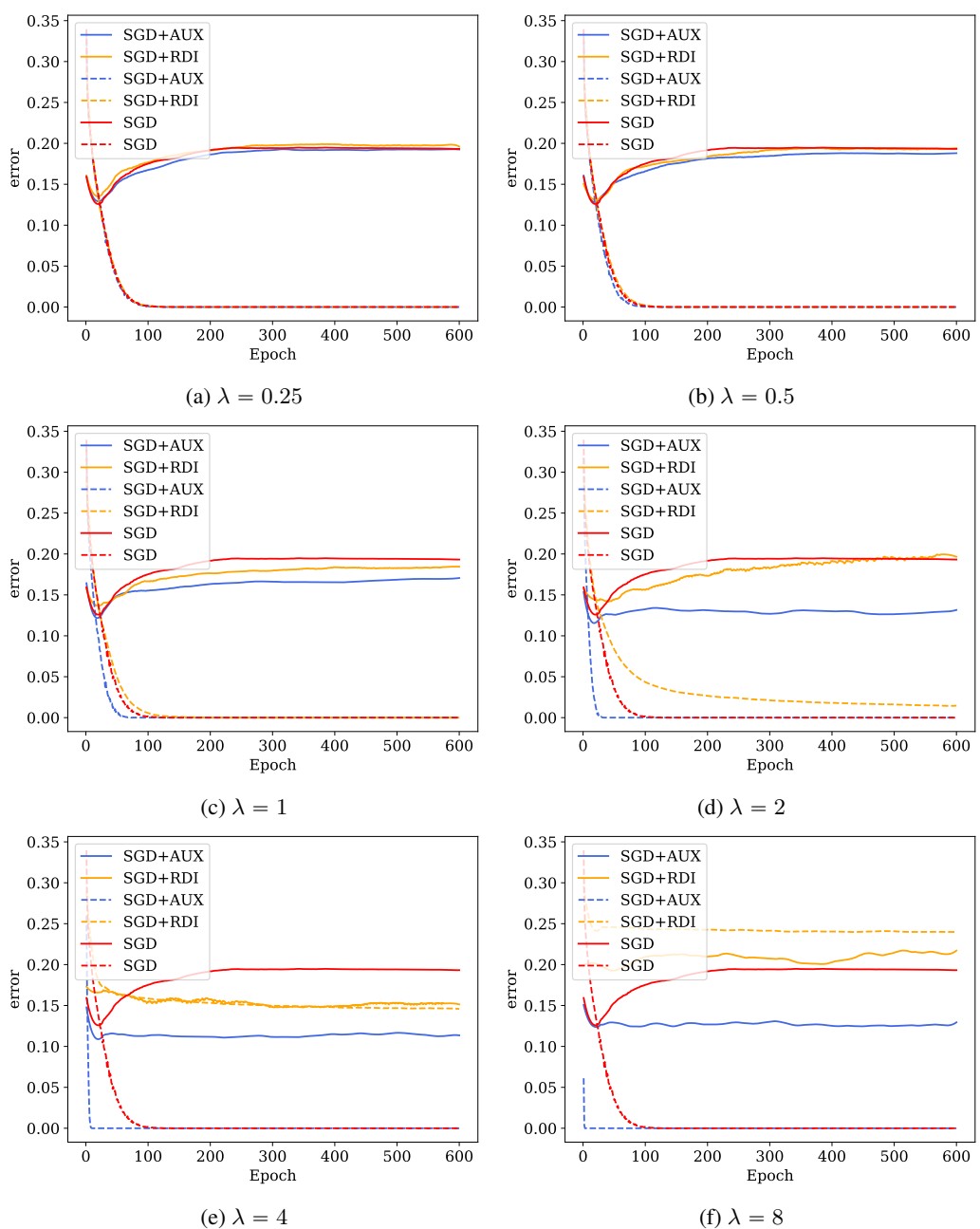

Figure 5: Training (dashed) & test (solid) errors vs. epoch for Setting 2. Noise rate = 20%, $\lambda \in \{0.25, 0.5, 1, 2, 4, 8\}$. Training error of AUX is measured with auxiliary variables.

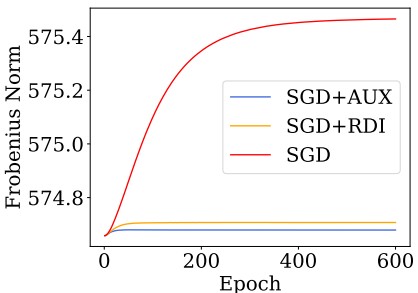 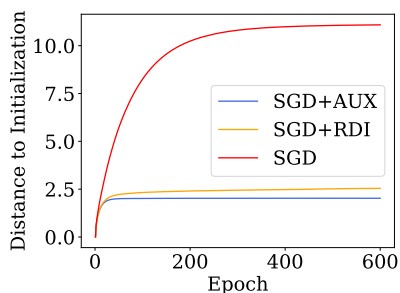

Figure 6: Setting 2, $\|\boldsymbol{W}^{(7)}\|_F$ and $\|\boldsymbol{W}^{(7)} - \boldsymbol{W}^{(7)}(0)\|_F$ during training. Noise rate $= 20\%$, $\lambda = 4$.

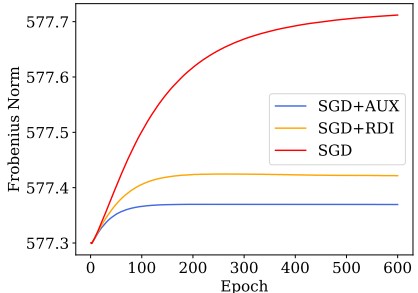 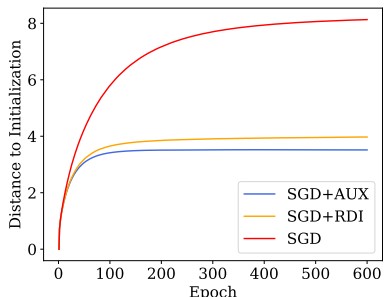

Figure 7: Setting 2, $\|\boldsymbol{W}^{(4)}\|_F$ and $\|\boldsymbol{W}^{(4)} - \boldsymbol{W}^{(4)}(0)\|_F$ during training. Noise rate $= 0$, $\lambda = 2$.

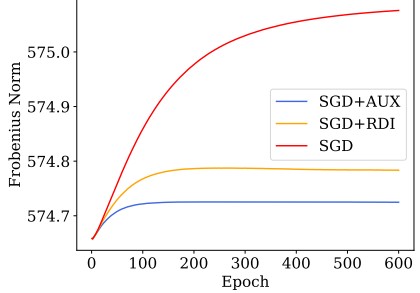 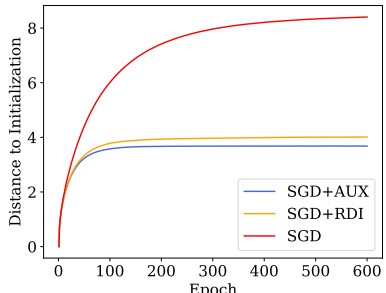

Figure 8: Setting 2, $\|\boldsymbol{W}^{(7)}\|_F$ and $\|\boldsymbol{W}^{(7)} - \boldsymbol{W}^{(7)}(0)\|_F$ during training. Noise rate $= 0$, $\lambda = 2$.

