# OpenReview forum: "Simple and Effective Regularization Methods for Training on Noisily Labeled Data with Generalization Guarantee"
_ICLR.cc/2020/Conference — Accept (Poster)_

### Official Review · AnonReviewer2 · 2019-10-18
**Official Blind Review #2**

**Rating:** 6

**Review:**

This paper studies the topic of learning with noisy labels, in particular, classification problem where the labels are randomly flipped with some probability. The main technical contributions of this paper are two folds: 1) proof of generalization bounds for kernel ridge regression solutions that depends on the clean labels only. 2) two regularization techniques that are shown to be equivalent to the kernel ridge regression when the neural networks approach the neural tangent kernel regime.

Numerical experiments are performed to verify that the proposed methods indeed helps over the baseline at fighting noisy labels. One weak point of the paper is that there is no comparison to any other methods designed to learn under noisy labels. Even though the paper states that the primary advantage of the proposed methods is simplicity, it would still be good to have some empirical comparison for reference.

I like that the paper has a section to explicit check whether the neural networks used in the experiments are in neural tangent kernel regime. However, I'm not very sure how to interpret the scale of the Frobenius norm in the change of the weights. Maybe one alternative approach is to compare the difference between the two neural tagent kernel --- one defined by theta(0), and one defined by theta(t) after training. Alternatively, maybe the experiments can be carried out with recent techniques to perform learning directly on infinite width networks (e.g. https://arxiv.org/abs/1904.11955 ).

**Experience Assessment:**

I have published in this field for several years.

**Review Assessment: Checking Correctness Of Derivations And Theory:**

I assessed the sensibility of the derivations and theory.

**Review Assessment: Checking Correctness Of Experiments:**

I assessed the sensibility of the experiments.

**Review Assessment: Thoroughness In Paper Reading:**

I read the paper at least twice and used my best judgement in assessing the paper.

---

> ### Author Response · Authors · 2019-11-15
> **Reply to Reviewer 2**
>
> Thank you for your valuable comments and for appreciating our work! Please see our response to each specific question below.
>
> --- Empirical comparison with other methods for noisy labels ---
> We compared our results with [Zhang and Sabuncu (2018)] which used the same ResNet-34 architecture, and we found that we can achieve better accuracy on noisy CIFAR-10. See Table 1 in the paper.
>
> In general we found it a bit difficult to have a fair comparison between different methods because different papers may use different network architectures. For example, the reported numbers in [Han et al. (2018)] are worse than ours on CIFAR-10, but they were using a simple 9-layer CNN instead of ResNet-34 that we used. For this reason we only wanted to make comparison with papers using similar architectures. Also as you mentioned the primary advantages of our methods are simplicity and generalization guarantee.
>
> [Zhang and Sabuncu (2018)] Generalized cross entropy loss for training deep neural networks with noisy labels. In NeurIPS 2018.
> [Han et al. (2018)] Co-teaching: Robust training of deep neural networks with extremely noisy labels. In NeurIPS 2018.
>
> --- Weight change v.s. kernel change ---
> In the NTK theory, the reason why the kernel doesn’t change much during training is because the weights in the network don’t change much. For a large width $m$, the relative change of weights $\frac{\|\theta(t)-\theta(0)\|}{\|\theta(0)\|}$ scales like $O(1/\sqrt{m})$. In Figure 3, we see that the relative change of weights in a particular layer is roughly $2/577=0.003$ which is tiny, an indication that the network is very likely in the NTK regime. We agree that comparing the kernels before and after training would also be useful in verifying that the network is in the NTK regime, and we plan to add this experiment to the final version of the paper. Thanks for the suggestion!
>
> --- Experiment directly on infinitely wide networks ---
> It would indeed be interesting to perform experiments on the exact NTK for infinitely wide networks, and this is exactly the setting our theory applies to. On the empirical side this approach may not achieve impressive results because: (i) the exact NTK computation on the entire CIFAR-10 dataset for CNN with pooling is very expensive, and is much more expensive than the standard training of a finite network; (ii) the performance of these kernels has reasonable test accuracy but is still not quite as good as trained finite networks - the best figure for clean CIFAR-10 from [Arora et al. (2019)] is only around 77%. Nevertheless, we think this experiment definitely has theoretical values, and we will consider including it in the final version if we obtain sufficient computation resources.

---

### Official Review · AnonReviewer3 · 2019-10-22
**Official Blind Review #3**

**Rating:** 8

**Review:**

In this paper, based on the effectiveness of early stopping in the training of noisily labeled samples, the authors proposed two intuitive (and novel) regularization methods: (1) regularizing using distance to initialization (2) adding an auxiliary variable b_i for every input x_i during training. In terms of theory, the authors showed that in the NTK regime, both regularization methods trained with gradient descent are equivalent to kernel ridge regression.  Moreover, the authors also provided a generalization bound of the solution on the clean data distribution when trained with noisy label, which was not addressed in previous research.

Overall, the paper is very well-organized and well-written. The contribution of the paper is significant, and numerical results also vindicate the theory developed in the manuscript. I recommend accepting the paper.

Two minor questions that are not going to affect my rating:
1. The theory developed in the paper depends highly on NTK. What if the network is not sufficiently wide (which is usually the case in practice), and the loss function is not L2?
2. In the second method using the auxiliary variable, it seems that every training sample x_i needs a variable b_i. Is this going to cause any problem in practice if data augmentation is used?


**Experience Assessment:**

I have read many papers in this area.

**Review Assessment: Checking Correctness Of Derivations And Theory:**

I assessed the sensibility of the derivations and theory.

**Review Assessment: Checking Correctness Of Experiments:**

I assessed the sensibility of the experiments.

**Review Assessment: Thoroughness In Paper Reading:**

I read the paper at least twice and used my best judgement in assessing the paper.

---

> ### Author Response · Authors · 2019-11-15
> **Reply to Reviewer 3**
>
> Thank you for your valuable comments and for appreciating our work! Please see our response to each specific question below.
>
> --- What if the network is not sufficiently wide, and the loss function is not L2? ---
> Our theoretical results don’t apply to networks that are not in the NTK regime or to other loss functions. Nevertheless, our experiments show that the proposed regularization methods are still very effective in these scenarios. We believe that a very interesting direction of future work is to understand this theoretically.
>
> --- Auxiliary variables with data augmentation? ---
> Auxiliary variables can indeed be used together with data augmentation. In our experiment on CIFAR-10, we used data augmentation and just used one auxiliary variable for each sample and its augmented samples. We will mention this clearly in the paper.

---

### Official Review · AnonReviewer1 · 2019-10-23
**Official Blind Review #1**

**Rating:** 6

**Review:**

This paper proposes two regularization methods for learning on noisily labeled data: the first penalizes the distance w.r.t. Euclidean norm from an initial point and the second uses an additional auxiliary variable for each example to learn a noise. In the theoretical part, the paper shows that an original clean dataset can be learned from a noisily labeled dataset based on NTK-theory. Finally, the effectiveness of proposed regularization methods is well verified empirically for 2-layer NN, CNN, and ResNet on image classification tasks (MNIST, CIFAR-10).

Contributions:
- Propose two simple regularization techniques for learning from a noisily labeled dataset.
- Give generalization guarantees for these methods

Clarity:
The paper is well organized and easy to read.

Quality:
The work is of good quality and is technically sound.

Significance:
Since proposed methods are in some sense related to the early-stopping for the (stochastic) gradient descent, the developed theory is useful in understanding the generalization ability of over-parameterized neural networks falling into NTK-regime. Although an additive noise setting for the regression problem is rather common in statistical learning theory, an artificially flipped label setting for the classification problem may be new except for a few recent studies [Li+(2019)]. A result (Theorem 5.1) for the regression problem with the squared loss is not so surprising because the generalization error of gradient descent in a high-dimensional space (e.g., over-parameterized NNs) or an RKHS (i.e., infinite-dimensional model) has been well studied and the generalization error is composed of the (constant) variance and the distance between the model output and the true label. Thus, existing results of generalization error analyses for the regression are directly applied to bound the prediction error for clean labels. However, I basically like this paper and I think this paper makes a certain contribution to understanding the effect of over-parameterization.

A few questions:
- Usually, the regularization parameter goes to zero as the number of examples increases. Conversely, the regularization parameter in the proposed methods also increases. Could you explain why this difference happens?
- In classification tasks, the original problem setting is recovered by setting $p=lamba=0$. However, the generalization bound by Theorem 5.3 is still affected by $\lambda$. Is this theorem tight?

-----
Update:
I thank the authors for the response. I have read the revised version of the paper and have confirmed an improvement of Theorem 5.2. I will keep my score. This paper is of good quality.

**Experience Assessment:**

I have read many papers in this area.

**Review Assessment: Checking Correctness Of Derivations And Theory:**

I assessed the sensibility of the derivations and theory.

**Review Assessment: Checking Correctness Of Experiments:**

I assessed the sensibility of the experiments.

**Review Assessment: Thoroughness In Paper Reading:**

I read the paper at least twice and used my best judgement in assessing the paper.

---

> ### Author Response · Authors · 2019-11-15
> **Reply to Reviewer 1**
>
> Thank you for your valuable comments and for appreciating our work! Please see our response to each specific question below.
>
> --- Why does the regularization parameter increase with the sample size? ---
> This is only due to a difference in normalization. In our definition the regularization is added to the sum of losses on all examples – see Eqn. (3). If we average the losses over all examples, the regularization parameter in Eqn. (3) becomes $\lambda^2 / n$ which decreases with the sample size $n$.
>
> --- Does Thm 5.2 apply to the noiseless case ($p=\lambda=0$)? ---
> Thank you for pointing out this issue! We have updated Thm 5.2 so that it can cover the case $p=0$ and $\lambda \to 0$. Basically there should be an additional factor of $\sqrt{p}$ in two terms in the bound. (Previously we simply upper bounded $p$ by $1$.) The last term still contains $\log \frac{n}{\delta \lambda}$ which is bad for small or zero $\lambda$, but the $\lambda$ there can actually be replaced with the minimum eigenvalue of the kernel matrix $k(X, X)$ so it is fine. Thm 5.3 can also be modified similarly.

---

### Decision · Program_Chairs · 2019-12-19

**Decision:**

Accept (Poster)

**Comment:**

This paper studies the effect of various regularization techniques for dealing with noisy labels. In particular the authors study various regularization techniques such as distance from initialization to mitigate this effect. The authors also provide theory in the NTK regime. All reviewers have positive assessment about the paper and think is clearly written with nice contributions but do raise some questions about novelty given that it mostly follows the normal NTK regime. I agree that the paper is nicely written and well-motivated. I do not think the theory developed here fully captures all the nuances of practical observations in this problem. In particular, with label noise this theory suggests that test performance is not dramatically affected by label noise when using regularization or early stopping where as in practice what has been observed (and even proven in some cases) is that the performance is completely unaffected with small label noise. I think this paper is a good addition to ICLR and therefore recommend acceptance but recommend the authors to more clearly articulate the above nuances and limitations of their theory in the final manuscript.